# Enhancer and super-enhancer dynamics in repair after ischemic acute kidney injury

Julia Wilflingseder [1,2,3,5✉], Michaela Willi [2,5], Hye Kyung Lee [2], Hannes Olauson [1,4], Jakub Jankowski[2,3], Takaharu Ichimura[1], Reinhold Erben [3], M. Todd Valerius [1], Lothar Hennighausen[2] & Joseph V. Bonventre [1✉]

The endogenous repair process can result in recovery after acute kidney injury (AKI) with adaptive proliferation of tubular epithelial cells, but repair can also lead to fibrosis and progressive kidney disease. There is currently limited knowledge about transcriptional regulators regulating these repair programs. Herein we establish the enhancer and super-enhancer landscape after AKI by ChIP-seq in uninjured and repairing kidneys on day two after ischemia reperfusion injury (IRI). We identify key transcription factors including HNF4A, GR, STAT3 and STAT5, which show specific binding at enhancer and super-enhancer sites, revealing enhancer dynamics and transcriptional changes during kidney repair. Loss of bromodomain-containing protein 4 function before IRI leads to impaired recovery after AKI and increased mortality. Our comprehensive analysis of epigenetic changes after kidney injury in vivo has the potential to identify targets for therapeutic intervention. Importantly, our data also call attention to potential caveats involved in use of BET inhibitors in patients at risk for AKI.

[1] Brigham and Women's Hospital, Renal Division, Harvard Medical School, 4 Blackfan Circle, Boston, MA 02115, USA. [2] Laboratory of Genetics and Physiology, NIDDK, NIH, 8 Center Dr, Bethesda, MD 20814, USA. [3] Department of Physiology and Pathophysiology, University of Veterinary Medicine, Veterinärplatz 1, 1210 Vienna, Austria. [4] Division of Renal Medicine, Department of Clinical Science, Intervention and Technology, Karolinska Institutet, Solnavägen 1, 171 77 Stockholm, Sweden. [5] These authors contributed equally: Julia Wilflingseder, Michaela Willi. ✉email: julia.wilflingseder@vetmeduni.ac.at; joseph_bonventre@hms.harvard.edu

Acute kidney injury (AKI) is common, present in up to 10% of hospitalized patients[1], up to 17.5% of patients with cancer, and is associated with high morbidity and mortality[2,3]. Common causes of AKI include ischemia reperfusion injury (IRI), sepsis, and exposure to nephrotoxic substances.

The physiological and cellular hallmarks of AKI include the loss of renal glomerular and tubular function, cell death of kidney epithelia cells, vasoconstriction, and initiation of an inflammatory response. AKI is defined clinically by a rise in serum creatinine reflecting the reduction of glomerular filtration. Although a decrease of renal function can be reversed through the endogenous repair processes of the kidney, AKI is associated with a substantially increased risk of developing fibrosis and chronic kidney disease (CKD), especially if there is severe or repeated injury resulting in 'maladaptive' repair processes[4,5].

Surviving tubular epithelial cells are the main cellular source of the repair process in the kidney with robust proliferation to replace the cells lost as a result of the injury[6]. In this process the proliferating tubular epithelial cells are able to rapidly activate transcriptional repair programs[7]. Investigation of gene expression and function have generated molecular models of kidney injury and repair[8]; yet, there is limited understanding of the epigenetic regulatory events that activate and regulate kidney tissue repair programs.

Recent advances in chromatin analyses suggest that gene regulatory elements are highly prevalent in the genome and, of these elements, distal-acting regulatory sequences, or enhancers, represent the most abundant class[9,10]. Enhancers can induce direct expression of their target genes by increasing transcription[11], and have been predominantly examined as a means for stage- and tissue-specific regulation during embryonic development[12,13]. Studies have also implicated enhancers in health and disease[14,15]. Such findings raise the possibility of existing enhancer elements that engage with transcription factors in response to tissue damage to regulate genetic programs for kidney repair.

To investigate the role of enhancer activation in kidney repair we made use of pharmacological enhancer inhibition through BET (bromodomain and extra terminal) inhibitors. BET protein family members couple with acetyl-lysine residues on histone tails[16] and interact with several key proteins involved in transcriptional regulation (initiation and elongation of transcription) at enhancer sites[17,18]. Bromodomain containing protein 4 (BRD4) is the most widely studied member of the BET protein family. Pharmacological inhibition of BET proteins shows therapeutic activity in a variety of different pathologies, particularly in models of cancer and inflammation[19–21]. Several of these BET inhibitors have been or are being evaluated in clinical trials with more than twenty studies currently registered at clinicaltrials.gov. Most studies evaluate the effect of BET inhibitors in different forms of cancer, but two trials are also testing this class of drugs for cardiovascular diseases and CKD (NCT02586155 and NCT03160430).

In this study, we demonstrate that kidney injury leads to genome-wide changes in the enhancer and super-enhancer repertoire with activation of injury responsive regulatory elements shaping the transcriptional response after injury and during repair. Further, we find that the transcription factors, hepatocyte nuclear factor 4 alpha (HNF4A), glucocorticoid receptor (GR), and signal transducer and activator of transcription (STAT) 3 and 5 bind at the identified enhancer elements. Inhibition of BRD4 function, and therefore blockage of enhancer dynamics after kidney injury, affects critical repair responses resulting in impaired recovery after AKI. These results are the first comprehensive characterization of enhancer and super-enhancer elements in the kidney in vivo before and after injury, thereby providing a rich resource for the research community and critical insight in the regulation of kidney repair.

## Results

**Kidney injury leads to enhancer repertoire changes in vivo.** In order to define the dynamic enhancer and super-enhancer landscape after kidney injury, we performed chromatin immunoprecipitation with next-generation DNA sequencing (ChIP-seq) in SHAM and bilateral-IRI samples at day two after surgery. We used day two post injury samples to capture the proliferating repair phase of the kidney[7]. Kidney cortex samples, consisting in a healthy kidney of at least 90% epithelia cells, were used to capture the chromatin state of mainly proximal (predominant) and distal tubular cells[22].

First, we established a chromatin landscape for the kidney using H3K4me3 to identify promoters, H3K27ac to reveal active enhancer regions and Pol II to define regions of active transcription. We then assessed the genome-wide localization of the representative BET protein, BRD4, also by ChIP-seq and performed sample matched RNA-seq in SHAM and IRI treated kidney samples (Fig. 1a).

H3K27ac marks strongly correlated with binding sites of BRD4 in both SHAM and IRI samples (Fig. 1b, Supplementary Fig. 1) indicating that both ChIP-seq profiles classify active enhancers in the kidney[23,24]. We applied broad peak calling on H3K27ac marks to identify active enhancer regions. Possible promoter regions were identified by H3K4me3 marks and H3K27ac peaks ±2500 bp from the transcription start site (TSS) of genes and verified by overlapping them. We identified 29,925 H3K27ac binding sites shared between SHAM and IRI samples, and 7,439 and 11,406 sites with preferential binding in SHAM or IRI, respectively (Fig. 1c). For further analysis, we named the three groups of binding sites: SHARED (present in both SHAM and IRI samples); IRI-decreased (preferential binding in SHAM samples); and IRI-increased (preferential binding in IRI samples) (Fig. 1d).

Amongst the SHARED sites, 56% are enhancers, and 44% (40% with H3K4me3 and 4% without) were identified as promoters (Fig. 1d, left). In both IRI- decreased and IRI-increased binding sites, the proportion of promoter to enhancers was very different compared to the SHARED group. The analysis of the peaks identified in the IRI-decreased group revealed 88% enhancers and only 12% promoters (2% showing H3K4me3 binding) (Fig. 1d, center). The peaks characteristic for IRI-increased sites consisted of 86% enhancers, and 14% promoters (7.5% with and 6.5% without H3K4me3 binding) (Fig. 1d, right), indicating a higher dynamic at enhancer sites compared to promoter sites after injury.

In order to verify the identified classes of enhancers and promoters we used peak profiles to show the underlying H3K27ac, BRD4, and Pol II marks for SHAM and IRI. Based on the ChIP-seq results for SHAM, the enhancers identified in the SHARED group show the highest coverage (solid orange line), followed by IRI-decreased (dashed black line). The enhancers present in IRI-increased are, as expected, the lowest (dotted red line; Supplementary Fig. 2a). Using IRI ChIP-seq samples, the enhancers gained in IRI (IRI-increased, dotted red line) have a higher coverage than those lost after injury (IRI-decreased, dashed black line). The SHARED enhancers are again those with the highest binding profile (solid orange line; Supplementary Fig. 2b). Based on these results, we conclude that the response to IRI leads to genome-wide alterations in the enhancer landscape in kidney epithelia cells.

**Gained enhancers regulate expression of injury-induced genes.** To determine the functional relevance of the identified enhancer

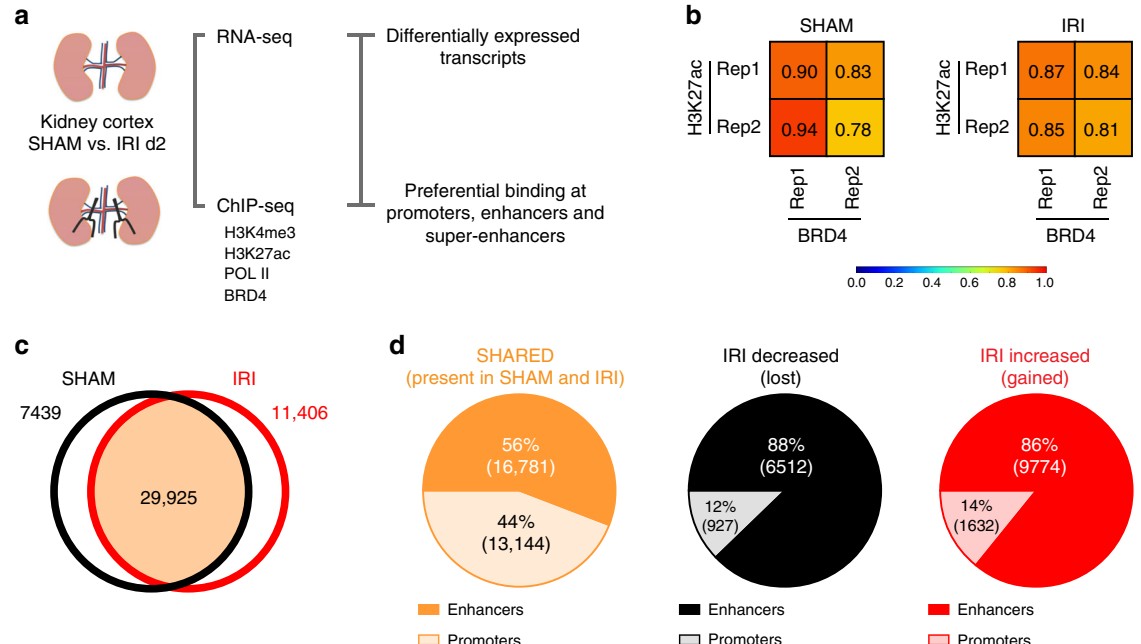

**Fig. 1 Identification of enhancer elements. a** Workflow of the experimental approach. **b** H3K27ac marks and BRD4 binding sites strongly correlate in both SHAM and IRI samples. **c** Broad peak calling was applied on H3K27ac SHAM and IRI samples. 29,925 peaks overlapped between them; 7439 and 11,406 were only identified in SHAM or IRI samples, respectively. **d** 56% of the identified peaks SHARED between both samples are enhancer elements, whereas 44% are promoter elements. Peaks identified with preferential binding in SHAM samples (IRI-decreased) consist of 88% enhancers and 12% promoter regions. Out of 11,406 peaks identified with preferential binding in IRI samples (IRI-increased) 86% are enhancers and 14% are promoters.

elements we leveraged sample-matched RNA-Seq gene expression data. The RNA-seq data revealed 919 significantly down-regulated genes and 1716 significantly upregulated genes when comparing IRI day 2 vs SHAM treated kidneys with false discovery rate (FDR) < 5% and fold change ±2 (Fig. 2a, Supplementary Data 1 and 2). It is interesting that the two most highly upregulated genes are *Lcn2 (NGAL)* and *Havcr1 (KIM-1)*, both of which are generally considered the two most highly upregulated genes after kidney injury[25,26].

Enhancers elements were assigned to genes using GREAT[27]. The majority of enhancers in each group were assigned to at least one gene (Fig. 2b). Only a few enhancers remained unannotated as they were more than 100 kb away from the nearby gene. The vast majority of enhancers were assigned to genes present within 5–50 kb (Fig. 2c).

We found that 571 enhancers were associated with 919 differentially downregulated genes (62%) and 1076 enhancers with 1716 upregulated genes (63%) in IRI day 2 vs SHAM treated kidneys (Fig. 2d). Overall, genes associated with increased enhancers showed significantly increased expression, and decreased enhancers showed statistically significant downregulation of the assigned genes, compared to genes associated with SHARED enhancers (Fig. 2e, Supplementary Data 3–5). As examples of putative enhancers at kidney related genes we show *Slc34a1* (which encodes the proximal tubule sodium-dependent phosphate transporter 2A) and *Kl* (klotho) locus. *Havcr1* genomic locus is shown as a kidney injury related gene. *Slc34a1* and *Kl* were significantly downregulated and *Havcr1 (KIM-1)* significantly upregulated after kidney injury on gene expression level (Fig. 2f) with corresponding protein level changes for KL and KIM-1 (Havcr1) (Supplementary Fig. 3). BRD4 and H3K27ac signals at enhancers near down-regulated *Slc34a1* and *Kl* (Fig. 2g, h) were significantly decreased after IRI, which is consistent with reduced promoter binding of H3K4me3 and significantly lower gene expression after an ischemic insult (Fig. 2f). In contrast, we found injury-induced enrichment of BRD4 and H3K27ac binding

at enhancers and promoter near upregulated *Havcr1* genes (Fig. 2i). *Havcr1 (KIM-1)* is expressed in injured proximal tubular cells[25], where it is involved in clearing apoptotic cells and cell debri from the tubular lumen. Therefore, it plays an important role in the repair phase of tubular cells[28], although prolonged expression of *KIM-1* leads to interstitial fibrosis and CKD[29].

These data suggest an association between BRD4 and H3K27ac enrichment at enhancers and the expression levels of their associated genes. Thus, the enhancer landscape changes in concert with the gene expression changes caused by kidney injury.

**Super-enhancer regulation in kidney repair.** Multiple and enriched H3K27ac/BRD4 signals in genomic proximity are coined super-enhancers and are associated with cell identity genes in normal and disease states[30,31]. To determine whether super-enhancers might play a role in the response to kidney injury, we undertook a systematic mapping of super-enhancers in IRI day 2 and in SHAM kidneys. We used H3K27ac and BRD4 co-occupied regions with preferential binding either in SHAM or IRI, or present in SHAM and IRI to identify super-enhancers using the ROSE algorithm[30,31]. This algorithm stitches enhancers present within 12.5 kb of each other and ranks them according to the BRD4 and H3K27ac signals (Fig. 3a). We identified 216 injury-associated super-enhancers consisting of 565 single enhancers, 164 super-enhancers lost during injury consisting of 419 enhancers and 385 SHARED super-enhancers consisting of 1630 enhancers, which are present in the kidney at baseline and in IRI day 2 samples. These results indicate widespread changes in H3K27ac and BRD4 occupancy at several super-enhancers after injury. Injury gained super-enhancers showed increased enrichment of H3K27ac and BRD4 at their different enhancer sites. By contrast, lost super-enhancers after IRI exhibited decreased enrichment of H3K27ac and BRD4 at their enhancer sites. The average profile of H3K27ac, BRD4 and Pol II is enriched over all

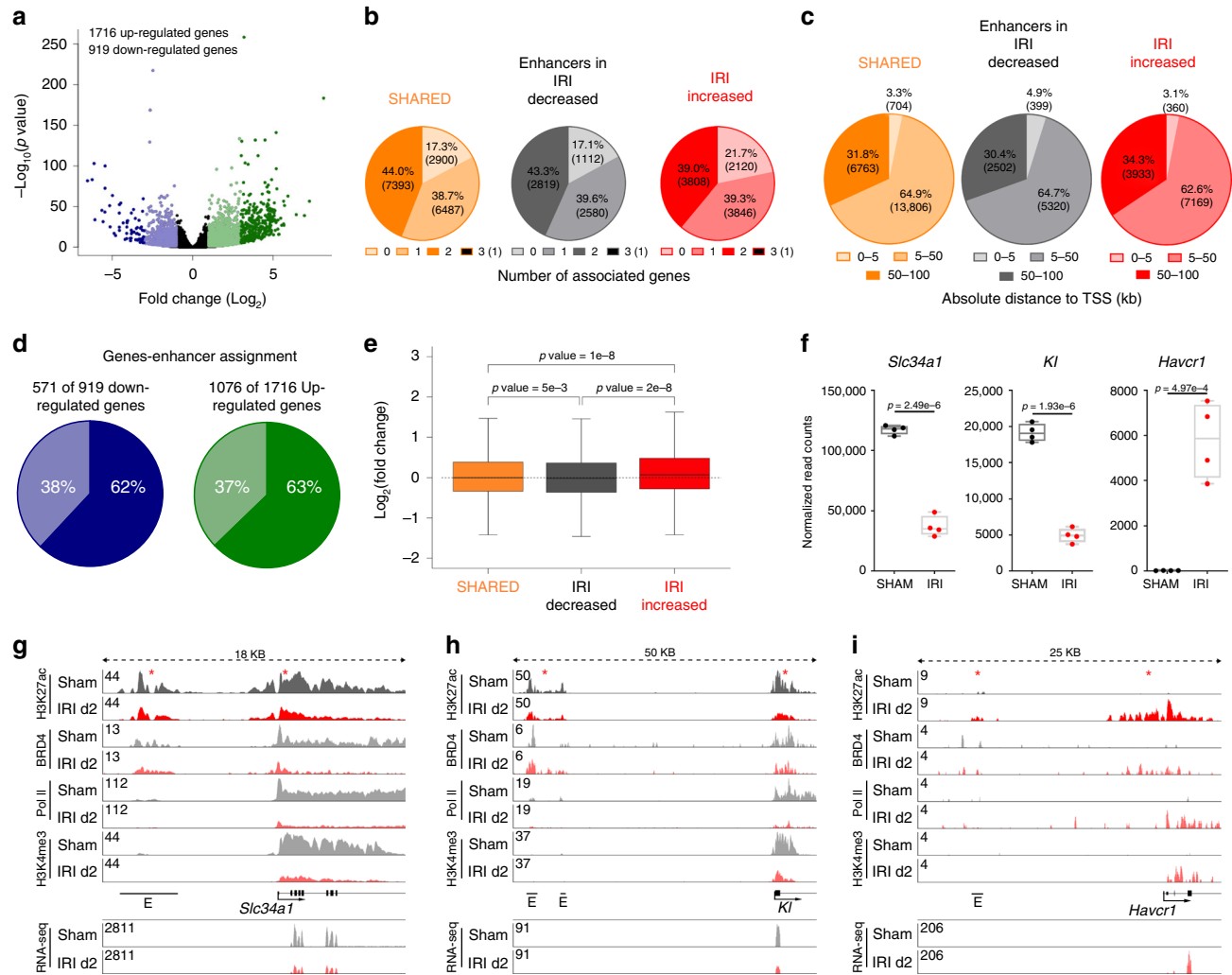

**Fig. 2 Enhancer—gene annotation. a** The volcano plot illustrates the 919 down- (blue) and 1716 upregulated genes (green) using RNA-seq data. **b** GREAT was used to annotate enhancers to genes. The pie plot shows how many genes were annotated to enhancer elements. **c** The pie plot illustrates the distance between the enhancers and their annotated genes (in kb to the transcriptional start site (TSS)). **d** 571 enhancer elements are annotated to the 919 downregulated genes (62%, dark blue). 1076 enhancers are assigned to 1716 upregulated genes (63%, dark green). **e** Genes associated with IRI-decreased enhancers ($n = 6512$) are significantly downregulated, whereas genes assigned to IRI-increased enhancers ($n = 9774$) have a significantly higher gene expression compared to genes assigned to SHARED enhancers ($n = 16781$). Median, middle line inside each box; IQR (interquartile range), the box containing 50% of the data; whiskers, 1.5 times the IQR. One-way ANOVA with Tukey post hoc test was applied. **f** *Slc34a1* and *Kl* are representative genes that were down-regulated in IRI day 2 when compared with SHAM. *Havcr1*(encoding KIM-1) is a gene significantly upregulated at IRI day 2. Two-sample t-test was applied (two-sided). Individual data points and box-plot with mean ± max, min are shown. $n = 4$ biologically independent samples **g** Illustration of the enhancer landscape of *Slc34a1*. The indicated enhancers show a lower coverage of H3K27ac, and a strongly reduced coverage of BRD4 and Pol II with IRI compared to SHAM. The coverage of BRD4, Pol II and H3K4me3 is also strongly reduced over the promoter with IRI. RNA-seq track shows reduced expression on exon elements after injury. **h** Depiction of the enhancers associated to *Kl*. These enhancers show a strong decrease of H3K27ac, BRD4, and Pol II. In addition the promoter loses H3K27ac, BRD4, Pol II, and H3K4me3 binding. RNA-seq track shows reduced expression on exon elements after injury. **i** *Havcr1* gains an enhancer element at IRI day 2. Also the promoter shows an increased coverage of H3K27ac, BRD4, Pol II, and H3K4me3. RNA-seq track shows increased expression on exon elements after injury.

identified super-enhancers compared to lone enhancers (Supplementary Fig. 4).

Integration with RNA-Seq data showed that the expression changes for super-enhancer associated genes were more pronounced than lone enhancer associated genes (Fig. 3b). Moreover, genes associated with gained super-enhancers had significantly higher fold changes in expression compared to genes associated with lost or unaffected super-enhancers (Fig. 3c). One example for a gene with increased expression and a gained super-enhancer is *Spp1* (Fig. 3d, e). The *Spp1* gene product had the highest abundance according to normalized read counts after kidney injury compared to all other genes. Also the Spp1 protein plasma

levels are significantly increased of this secretory protein (Supplementary Fig. 3c). *Hnf1b*, an important transcription factor in the kidney epithelium, which controls expression of genes responsible for development, solute transport and metabolism, is a representative gene for the class of unchanged genes and of SHARED super-enhancer where there is no change with IRI (Fig. 3d, f, Supplementary Fig. 3)[32]. Overall, 83% of the identified super-enhancers (SHARED combined with IRI-decreased super-enhancers in SHAM samples) and 71% of SHARED combined with IRI-increased super-enhancers in IRI samples can be annotated to at least one highly abundant gene (top 20%) in SHAM and IRI day 2 samples, respectively

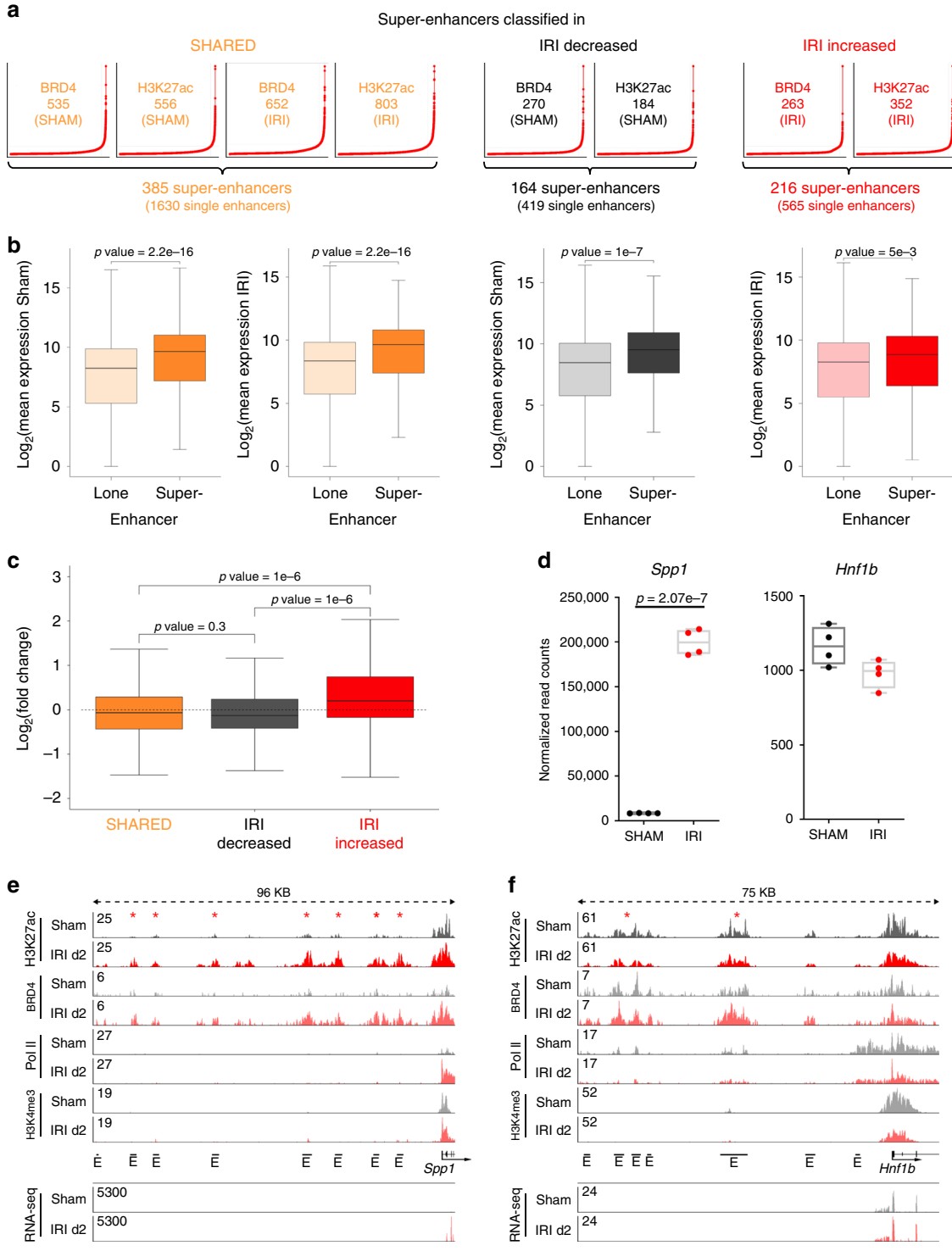

(Supplementary Data 6–8), reflecting super-enhancer association to high abundant gene expression and the important role of super-enhancers in kidney epithelial cells.

**Transcription factor binding predicted by motif analysis.** Little is known about the characteristics of transcription factor binding at specific enhancer sites in the regulation of repair. Therefore and to further validate the identified enhancer and super-enhancer sites we studied potential transcription factor (TF) binding motifs in our enhancer repertoire and subsequently measured a selection of predicted TFs with ChIP-seq.

First, we identified candidate TFs regulating the enhancer landscape in kidney cells via motif discovery analysis. The identified transcription factor motifs in the SHARED, IRI-decreased, and IRI-increased enhancer groups are shown in Supplementary Data 9. Further, Supplementary Fig. 5a shows the top ten transcription factor motifs in the SHARED, IRI-decreased and IRI-increased enhancer groups as defined by $p$-value. The $p$-value reflects the enrichment of the TF motif in the enhancer sequences compared to the background sequences. In contrast, Supplementary Fig. 5b shows the percentage of enhancer sequences having the underlying TF motif. Among the predicted

**Fig. 3 Identification of super-enhancers and their annotation to genes. a** Super-enhancers were identified for all three enhancer categories using H3K27ac enhancer elements with the underlying H3K27ac and BRD4 signal. The resulting individual super-enhancers using H3K27ac or BRD4 were overlapped to identify the 385 super-enhancers based on enhancers identified in SHARED, the 164 super-enhancers based on IRI-decreased enhancers and 216 super-enhancers identified in IRI upregulated enhancers. **b** The box plots depict significantly higher expression for genes associated with super-enhancers. Unpaired $t$-tests were applied (two-sided). SHARED lone enhancer: $n = 15151$, SHARED super-enhancer: $n = 385$; IRI-decreased lone enhancer: $n = 6093$, IRI-decreased super-enhancer: $n = 164$; IRI-increased lone enhancer: $n = 9209$, IRI-increased super-enhancer: $n = 216$. Median, middle line inside each box; IQR (interquartile range), the box containing 50% of the data; whiskers, 1.5 times the IQR. **c** The box plot shows that genes assigned to super-enhancers from the IRI-decreased group ($n = 164$) have a lower gene expression level, whereas genes assigned to IRI- increased super-enhancers ($n = 216$) have significantly higher gene expression level compared to SHARED super-enhancers ($n = 385$). Median, middle line inside each box; IQR (interquartile range), the box containing 50% of the data; whiskers, 1.5 times the IQR. One-way ANOVA with Tukey post hoc test was applied. **d** *Spp1* as a representative gene for gaining transcriptional marks is significantly upregulated at IRI d2 (based on RNA-seq data). *Hnf1b* is a representative gene assigned to a SHARED super-enhancer. Two-sample $t$-test was applied (two-sided). Individual data points and box-plot with mean ± max, min are shown. $n = 4$ biologically independent samples. **e** The induced expression of *Spp1* goes along with gained enhancer elements at IRI d2. The enhancers are characterized by strongly increased H3K27ac and BRD4 coverage. The gene body shows strongly increased Pol II binding. RNA-seq track shows increased expression on exon elements after injury. **f** *Hnf1b* enhancers show mild reductions in BRD4, Pol II and H3K27ac binding. RNA-seq track shows unchanged expression on exon elements after injury.

transcription factors many of them are also differentially regulated after injury (Supplementary Fig. 5c).

Based on the motif discovery analysis and availability of suitable antibodies we selected transcription factors in each group for validation in ChIP-seq experiments. Out of the selected transcription factors, namely HNF4A[33,34], GR, Fos-related antigen 1 (FRA1), Fos-related antigen 2 (FOSL2), Jun proto-oncogne (JUN), and STAT 3 and 5 we were able to generate high quality ChIP-seq profiles for four transcription factors which have important roles in kidney epithelial cell fate and response to injury- HNF4A[33], GR[35], STAT3[36], and STAT5[37]. We analyzed the genome-wide binding of these four transcription factors on the identified enhancer elements in the three categories: SHARED, IRI-decreased, and IRI-increased (Fig. 4a). We observed reduced binding of HNF4A and GR at SHARED and IRI-decreased enhancer elements. STAT3 showed decreased binding at IRI-decreased enhancer elements and increased binding at IRI-increased enhancer elements. We could not observe a dynamic binding pattern of STAT5 after IRI. Representative enhancer loci with binding of the four transcription factors are shown in Fig. 4b and Supplementary Fig. 6. At the enhancer element next to Slc34a1 we observed reduced binding of HNF4A, GR, and STAT3. At the Junb locus we find additional STAT3 binding after IRI indicating Junb activation by STAT3 (Fig. 4b). Interestingly at the regulatory element next to Havcr1 we observed GR binding before injury and reduced GR binding after IRI indicating GR may have a repressive function on Havcr1 before injury (Supplementary Fig. 6a). Super-enhancer elements occupied by the identified transcription factors are shown in the Spp1 and Hnf1b locus (Supplementary Fig. 6a). Bcl6 and Neat1 control loci provide visual proof of the technical quality of the ChIP-seq in SHAM and IRI kidney cortex samples (Supplementary Fig. 6b).

In summary, we predicted distinct transcription factor panels and demonstrate three transcription factors, HNF4a, GR, and STAT3, which are potentially involved in enhancer dynamics and therefore potentially significantly contribute to shaping the enhancer landscape in the kidney before and after kidney injury.

**BET inhibition increases mortality after experimental AKI**. As proof of principle of the role of the observed enhancer and super-enhancer dynamic after kidney injury, we selectively disrupted the BET family, and examined the phenotypic consequences of BET dependent enhancer inhibition. The BET inhibitor, JQ1, was administered prior to injury or later during recovery from IRI in mice. First, we clarified the importance of the different BET family members (BRD2, BRD3, and BRD4) in the kidney. We

performed immunofluorescence staining and ChIP-seq experiments for BRD4, BRD2, and BRD3. We found that BRD4 is the dominant member of the BET family in the kidney with higher protein abundance and genome-wide binding at regulatory elements compared to BRD2 and BRD3 which have low protein abundance and low binding at the genome before and after IRI (Supplementary Fig. 7).

Wild-type adult male C57BL/6 mice were injected with JQ1 (50 mg/kg BW) daily starting at the day of IRI surgery (day 0) (Fig. 5a), a dose and schedule well tolerated in the previous studies[31,38,39]. Compared to SHAM mice we detected a 10 to 20-fold increase in serum creatinine at day 1 after IRI, indicating we sucessfully induced AKI in our IRI animals (Fig. 5b). The increase of serum creatinine at day 1 after IRI was equal between IRI vehicle and IRI JQ1 group, suggesting BET inhibition did not modify the ischemia-induced initial renal injury (Fig. 5b). Strikingly, while vehicle-treated mice showed the expected 20% mortality after 26 min bilateral warm ischemia, mice treated with JQ1 had 80% mortality dying 2–3 days post surgery (Fig. 5c, log rank $p = 0.0186$), indicating kidneys exposed to JQ1 treatment have an impaired recovery response after injury. Of note, surviving mice in the IRI JQ1 group were the ones with the lowest creatinine at day 1 post surgery.

To investigate whether the effects seen were specific to BET inhibition rather than nonspecifically induced by transcriptional inhibition we compared the response to inhibition of BETs to the response to CDK9 inhibition with flavopiridol. It was shown recently that BET proteins act as master transcription elongation factors independent of CDK9 recruitment[40]. We observed no significant increase in mortality with two tested concentrations of CDK9 inhibitor (flavopiridol 2.5 mg/kg and 1 mg/kg body weight per day when we started the treatment 3 h before the surgery (IRI SHAM: 75% survival, $n = 4$; IRI flavopiridol 1 mg/kg: 83% survival, $n = 6$; IRI flavopiridol 2.5 mg/kg: 67% survival, $n = 6$)).

To evaluate the effect of delayed treatment of JQ1 after AKI in mice and to estimate the time window in which JQ1 had a detrimental effect on the survival after AKI we treated C57BL/6 mice with JQ1 starting at day 1, day 2, day 3 or day 4 after IRI surgery (Fig. 5d). Again, serum creatinine at day 1 after injury was not different between the vehicle and JQ1 treatment groups (Fig. 5e). There was a significant increase in mortality rate after IRI at 83% (log rank $p = 0.0096$) in mice exposed to delayed JQ1 treatment starting at day 1. When JQ1 treatment was started at day 2 there was a higher mortality rate (50%), but without reaching statistical significance. When JQ1 was initiated on day 3 or 4 post IRI the survival rate did not change compared to the vehicle-treated group (Fig. 5f). The higher mortality rate when

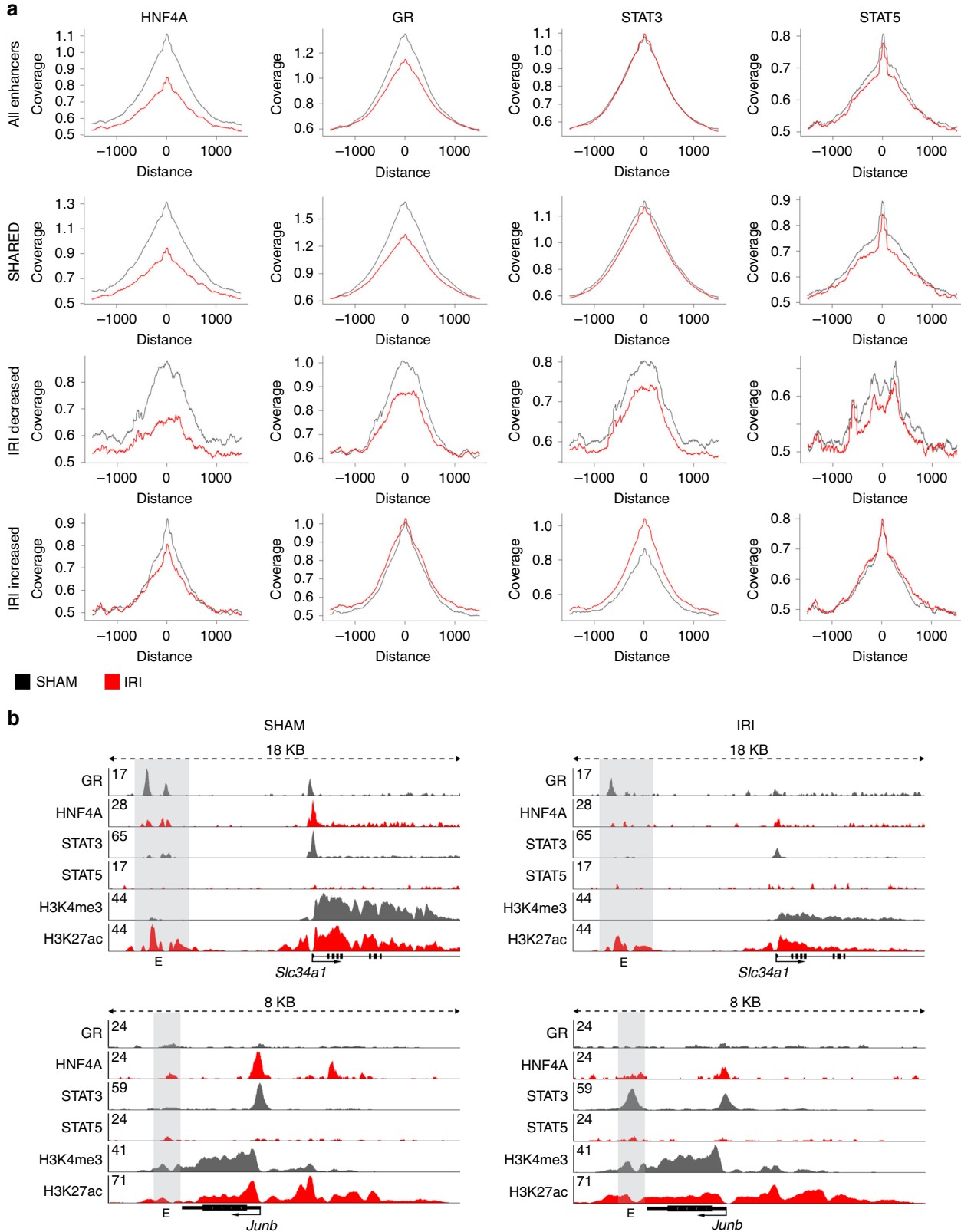

**Fig. 4 HNF4A, GR, STAT3, and STAT5 binding at enhancer elements. a** Genome-wide coverage plots at all, SHARED, IRI-decreased and IRI-increased enhancers. The coverage for HNF4A and GR decreased in SHARED and IRI-decreased enhancer group after IRI. In contrast, STAT3 shows increased coverage at IRI-increased enhancers. STAT5 peak height at enhancer elements is unchanged after kidney injury. **b** Representative examples of transcription factor binding at enhancer sites in kidney epithelia cells. The Slc34a1 and Junb genomic locus are shown for HNF4A, GR, STAT3, and STAT5 binding together with H3K4me3 and H3K27ac in the SHAM (left) and IRI (right) condition. Enhancer elements are indicated by gray bars.

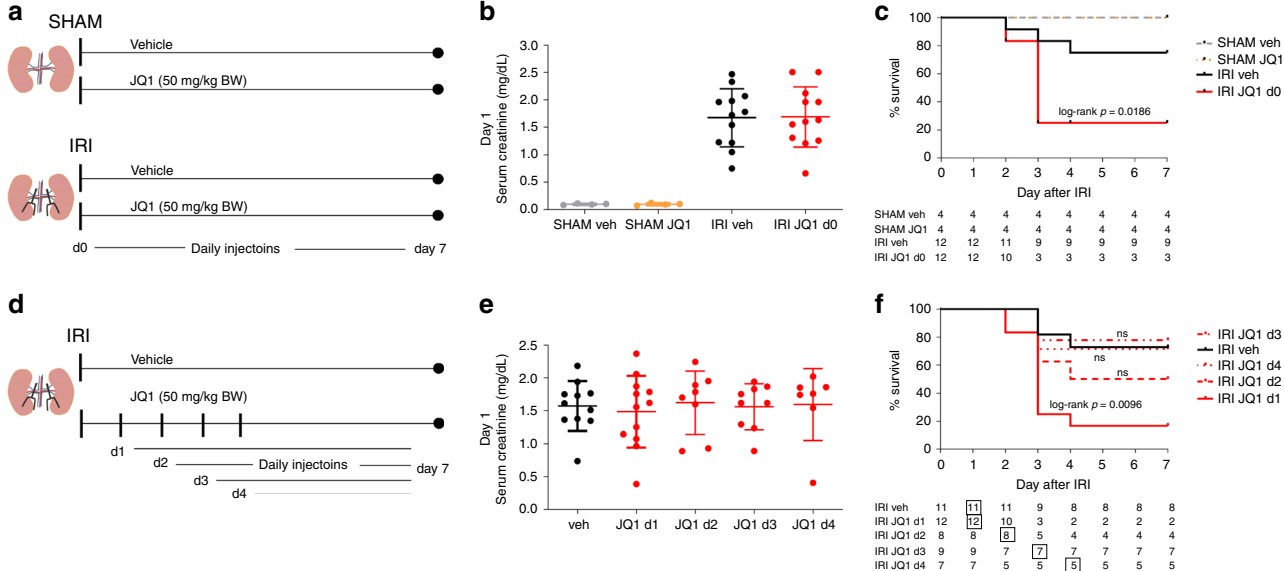

**Fig. 5 Phenotypic consequences of BET inhibition after experimental AKI. a** C57BL/6N mice (10- to 12-week-old males) were treated daily starting at the day of SHAM or IRI surgery (d0) with JQ1 (50 mg/kg BW) or vehicle (DMSO/10% ß-cyclo dextrin 1:10). **b** Serum creatinine (mg/dL) values in SHAM and IRI animals with vehicle or JQ1 at day 1 after surgery are shown (individual data points and mean ± SD). SHAM: $n = 4$; IRI: $n = 12$ biologically independent samples. **c** Survival curves after IRI surgery (ischemic time was 26 min at 37 °C). JQ1 treatment leads to 80% mortality after IRI surgery. Table shows number of animals alive at indicated days. **d** Experimental design of delayed treatment with JQ1 (50 mg/kg BW) after IRI starting at day 1, day 2, day 3 or day 4. **e** Serum creatinine levels at day 1 after IRI, verifying that AKI was induced to a roughly equivalent extent in each of the groups of animals that were subsequently treated with JQ1 (individual data points and mean ± SD). IRI veh: $n = 11$; IRI d1: $n = 12$; IRI d2: $n = 8$; IRI d3: $n = 9$; IRI d4: $n = 7$ biologically independent samples. **f** Survival curve after delayed JQ1 treatment starting day 1, day 2, day 3 or day 4. Table indicates number of animals alive at each day. Box indicates start of vehicle or JQ1 treatment. Source data are provided as a Source data file.

mice are treated with JQ1 within 48 h of IRI suggests that within this window of time BET dependent transcriptional activation is essential to control gene expression central to the early survival after IRI-induced AKI. The lack of mortality with JQ1 administration at later times after the initial ischemia also points to the lack of generalized toxicity and is compatible with an inhibitory effect on early repair as an explanation for the mortality seen with early JQ1 exposure after injury.

**Transcriptional consequences of BET inhibition**. To gain insights into the transcriptional consequences of BET inhibition in kidney injury and repair, we performed transcriptional profiling of vehicle- and JQ1-treated kidney cortex samples from animals at day 2 after IRI. Principal component analysis (PCA) shows clear separation between SHAM, IRI, and IRI JQ1 (d0) treated kidney profiles by visualizing principal component (PC) 1 (67% variance) and PC 2 (29% variance) (Fig. 6a). Also at day 2 after IRI we couldn't detect any significant difference in serum creatinine with JQ1 d0 treatment compared to the vehicle group (Fig. 6b). By contrast, at the transcriptional level a total of 2635 transcripts (1716 up, 919 down) were differentially regulated between SHAM vs. IRI and 3054 transcripts (676 up, 2378 down) were differentially regulated between IRI vs IRI JQ1 (Supplementary Data 1, 2, 10, and 11). 73% of downregulated genes (fold change > 2) after JQ1 treatment can be assigned to active enhancer elements (Chi-square: $p < 0.001$). Of the 1716 upregulated transcripts after IRI (Fig. 6c), 718 genes showed reduced expression if JQ1 treatment accompanied the IRI (Chi-square $p < 0.001$) (Fig. 6d), which means that approximately 40% of injury-induced genes were suppressed by JQ1 (Fig. 6c). JQ1 suppressed genes are associated with pathways central in kidney repair such as cell cycle, TNF-alpha signaling, ECM-receptor interaction, chemokine signaling, NOD-like receptor signaling, protein digestion and absorption, DNA replication, cytokine–cytokine

receptor interaction, PI3K-Akt signaling and p53 signaling pathway (Fig. 6e)[8,29,41,42]. The top 30 upregulated genes with the highest fold changes after kidney injury are shown in Fig. 6f. Out of the 30 genes 23 were suppressed by JQ1 treatment (FDR < 5%, at least 2-fold reduction), with many of the suppressed genes central to kidney repair such as *Cdk1* (Cyclin-dependent kinase involved in cell cycle progression)[6]. Also at the protein level JQ1 leads to a reduced expression of *KIM-1* and to a marked reduction of proliferating (Ki67+) cells (Fig. 6g). Corresponding transcript expression levels of *KIM-1* and *Mki67* are significantly reduced as validated by qRT-PCR (Fig. 6h).

To further illustrate the consequence of BET inhibition we performed Pol II ChIP-seq with JQ1 treated kidneys after IRI. Through competitive binding of the two extra terminal bromodomains of the BET proteins, JQ1 inhibits the transcriptional elongation process resulting in genome-wide Pol II pausing[43]. We found that JQ1 treatment leads to higher coverage of Pol II on the TSS (0% distance) and lower coverage in the middle of the gene body (50% distance) indicating genome-wide Pol II pausing two days after IRI in the kidney (Fig. 6i). Further, we show the Spp1 gene body for Pol II binding in SHAM, IRI, and IRI + JQ1 treatment (Fig. 6j). Spp1 mRNA expression was also downregulated by JQ1 (Fig. 6k). Together, these results clearly illustrate that BET inhibition impairs the transcriptional response to injury and disrupts acute repair programs in the kidney after IRI.

To also explore the function of BET proteins in the maintenance of kidney specific transcriptional programs without injury we compared SHAM vehicle vs SHAM JQ1 d0 treated kidneys at day 2 after SHAM surgery. Surprisingly, also in the healthy kidney BET inhibition leads to significant changes on the transcriptional level. A total of 2441 transcripts (774 up, 1667 down) were differentially regulated comparing SHAM vehicle vs. SHAM JQ1 (Supplementary Data 12 and 13). Among the 1667

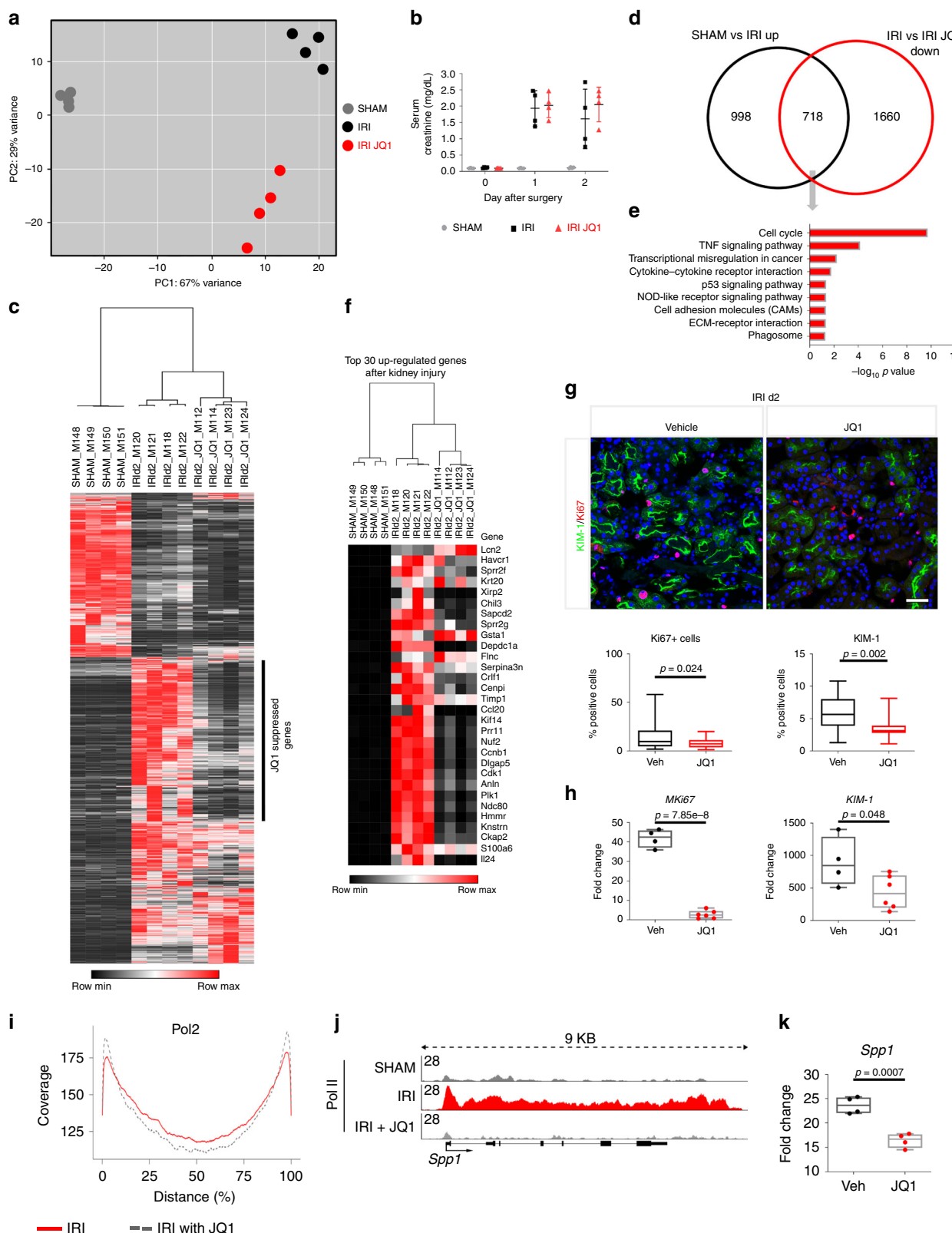

downregulated transcripts many genes are involved in the normal functions of the kidney such as genes with kidney specific transporter activity (Supplementary Data 14). Of the 334 transcripts identified as kidney related in a whole-body tissue comparision[44] 99 were significant down-regulated by JQ1 in the kidney (Supplementary Data 15, Chi-square $p < 0.001$). Therefore, our data suggest that BET proteins are important in the

maintenance of kidney specific transcriptional programs and essential in the transcriptional regulation of early kidney repair.

**Evaluation of BET inhibition in kidney fibrosis.** To evaluate kidney fibrosis development after IRI we treated C57BL/6 mice with JQ1 starting at day 4 after injury once the initial phase of post-ischemic transcriptional response had subsided[4]. Mice were

**Fig. 6 Transcriptional consequences of sBET Inhibition. a** Principal component (PC) analysis of normalized RNA-seq data matrix of SHAM, IRI and IRI JQ1 kidney cortex samples on day 2 after IRI. **b** Serum creatinine values at days 0, 1, and 2 after IRI comparing SHAM, IRI and IRI JQ1-treated groups. **c** Genes significantly differentially regulated between SHAM and IRI are shown in a heatmap of SHAM, IRI, and IRI JQ1; JQ1 leads to suppression of ~40% upregulated genes after injury. **d** Comparison of significantly upregulated genes 2 days after injury (SHAM vs. IRI up) and significantly down-regulated genes after JQ1 treatment (IRI vs IRI JQ1 down) with an overlap of 718 transcripts (Chi-square test $P < 0.001$) shown in a Venn diagram. **e** KEGG pathways enriched for the 718 genes upregulated after injury and down-regulated by JQ1. **f** Top 30 upregulated genes after injury shown in a heatmap of SHAM, IRI and IRI JQ1 samples. **g** Representative KIM-1/Ki67-immunostained IRI kidney cortex treated with vehicle or JQ1 at day 2 after injury, Quantification of Ki67+ cells (Ki67+ cells/total number of cells (DAPI)) and KIM-1+ surface area per hpf ($n = 4$, 7 high power fields (hpf) per sample). Scale bar: 50 μm. **h** Fold change of Mki67 and KIM-1 after IRI comparing vehicle and JQ1 treated animals at day 2 after injury (vehicle: $n = 4$, JQ1: $n = 6$). **i** Genome-wide assessment of Pol II binding: Genome-wide coverage blots of Pol II on the gene body. Pol II binding after JQ1 treatment is increased at the TSS and decreased across the gene body indicating Pol II pausing **j** Pol II ChIP-seq tracks at the Spp1 gene body. **k** Fold change of Spp1 after IRI comparing vehicle and JQ1 treated animals at day 2 after injury. $n = 4$. Data represent the mean ± min, max. Box contains 50% of the data. $t$-test (two-sided) (**g**, **h**, **k**). Source data are provided as a Source data file.

randomly assigned to vehicle or JQ1-treated groups (Supplementary Fig. 8a). Two mice died before the start of the treatment at day 3 after IRI. BUN trajectories were not different between vehicle and JQ1 group after injury (Supplementary Fig. 8b). Interestingly, also gene expression levels at day 21 of fibrosis-associated genes, such as KIM-1, Acta2, Ctgf, Col1a1, and c-Myc were not differentially regulated with JQ1 treatment (Supplementary Fig. 8c). Furthermore, we evaluated the amount of fibrosis in these kidneys by Masson Trichrome-staining at day 21 and saw no significant reduction by JQ1 treatment (Supplementary Fig. 8d). Also of note, the severity of fibrosis seen at day 21 after AKI induced by IRI was moderate with only ~5% Masson Trichrome-positive area in the vehicle group; therefore, we evaluated the effect of fibrosis development under JQ1 treatment in two different kidney fibrosis models where the fibrosis is more severe, namely aristolochic acid toxic nephropathy (AAN) and unilateral ureteral obstruction (UUO).

A single injection of aristolochic acid (AA) (3 mg/kg BW) to BALB/c mice was combined with a daily treatment of JQ1 in two treatment arms: (1) starting the same day as the AA injection (group: JQ1 d0) or (2) starting on day 7 after AA injection (group: JQ1 d7) (Fig. 7a). AA treatment alone resulted in severe kidney injury, as indicated by increased serum creatinine levels (Fig. 7b), but was not associated with mortality as there was 100% survival in the vehicle-treated mice (Fig. 7c). Daily JQ1 treatment commencing on day 0 or 7 after AA injection did not alter serum creatinine (Fig. 7b). Thirty-three percent mortality was detected in the early JQ1 treated mice (JQ1 d0; two out of six mice died, log rank $p = 0.072$), compared to no mortality in the delayed JQ1 treated mice (JQ1 d7). At the protein level JQ1 d0 and JQ1 d7 compared to vehicle lead to similar reductions of Acta2, and Ki67+ cells (Fig. 7d, e) on day 21, confirming reduced myelofibroblasts and proliferation in these kidneys.

To further evaluate the anti-fibrotic effects of JQ1, permanent injury was induced to the left kidney, through irreversible obstruction of the ureter resulting in a severe fibrotic phenotype within 10 days after obstruction (Fig. 7f). BET inhibition significantly reduced the severity of fibrosis in JQ1 treated C57BL/6N mice after UUO as reflected by decreased trichome-positive area and collagen deposition (assessed by Sirius red-staining) (Fig. 7g, h) which is consistent with the current literature[45–47]. No mortality was observed in the treatment arms as kidney function was maintained by the contralateral, uninjured kidney. In summary, these data derived from IRI, UUO and AAN models suggest that BET dependent gene regulation is essential in kidney repair and also plays a part in the development of kidney fibrosis post injury.

## Discussion

Surviving kidney tubular cells have to undergo phenotypic and transcriptional changes to initiate the repair of the tubular structure and restore kidney function after AKI. We have determined epigenetic changes that occur after kidney injury during the early repair phase. We profiled the enhancer and super-enhancer repertoire in normal kidney and during the proliferating repair phase 2 days after IRI. This yields a rich resource of epigenomic data of distal regulatory elements controlling the expression of critical genes and pathways involved in kidney injury and repair. Our data suggest that enhancer regulation and genes controlled by these enhancer elements are disproportionally enriched during repair. Early after an insult to the kidney, a cascade of downstream signaling events, involving cytokine, chemokine and kinase-dependent signaling[48], are activated[4]. These signals recruit key signal-dependent transcription factors, such as STAT3 or AP-1 transcription factors, mainly to distal regulatory enhancer and super-enhancer elements that further lead to histone acetylation and binding of mediator proteins, including BRD4. This transcriptional regulation cascade promotes transcription of genes key to kidney repair. We also observe a loss of enhancer and super-enhancer elements with injury reflected by decreased H3K27ac and BRD4 marks. We observed reduced binding of HNF4A and GR at IRI-decreased enhancer elements, changes that likely facilitate the dedifferentiation process necessary for surviving kidney cells to proliferate[42].

How transcription factors involved in kidney repair act on enhancer elements and their target genes in kidney cells is unknown[42]. Our motif discovery analyses within enhancer and super-enhancer elements provides insight into which transcription factors might contribute to the transcriptional initiation of kidney repair and maintaining kidney specific gene expression programs. The HNF4A transcription factor was the top predicted motif in our SHARED and IRI-decreased enhancer group suggesting enhancer regulation in kidney epithelia cells. In humans heterozygous mutations in this transcription factor causes variable multisystem phenotypes including maturity-onset diabetes of the young and a variety of renal phenotypes (including renal cysts and mimicking of Fanconi syndrome)[49]. This attests to the importance of HNF4A in the transcriptional regulation of kidney epithelia-specific genes. However, as HNF4A is not kidney specific and also has important roles in other organs such as the pancreas and the liver, it is likely that a panel of transcription factors are necessary for the regulation of kidney epithelia-specific transcription programs. We provide the first evidence that HNF4A and GR are part of such a transcription factor panel providing insight into the differentiation process of the kidney and therefore important information for regeneration medicine.

Related to the differentiation process into kidney epithelia cells, kidney repair requires specific transcriptional changes of surviving kidney epithelia cells that allows them to proliferate. We found a dynamic binding of STAT3 at injury regulated enhancer elements. We show an interesting relationship with STAT3

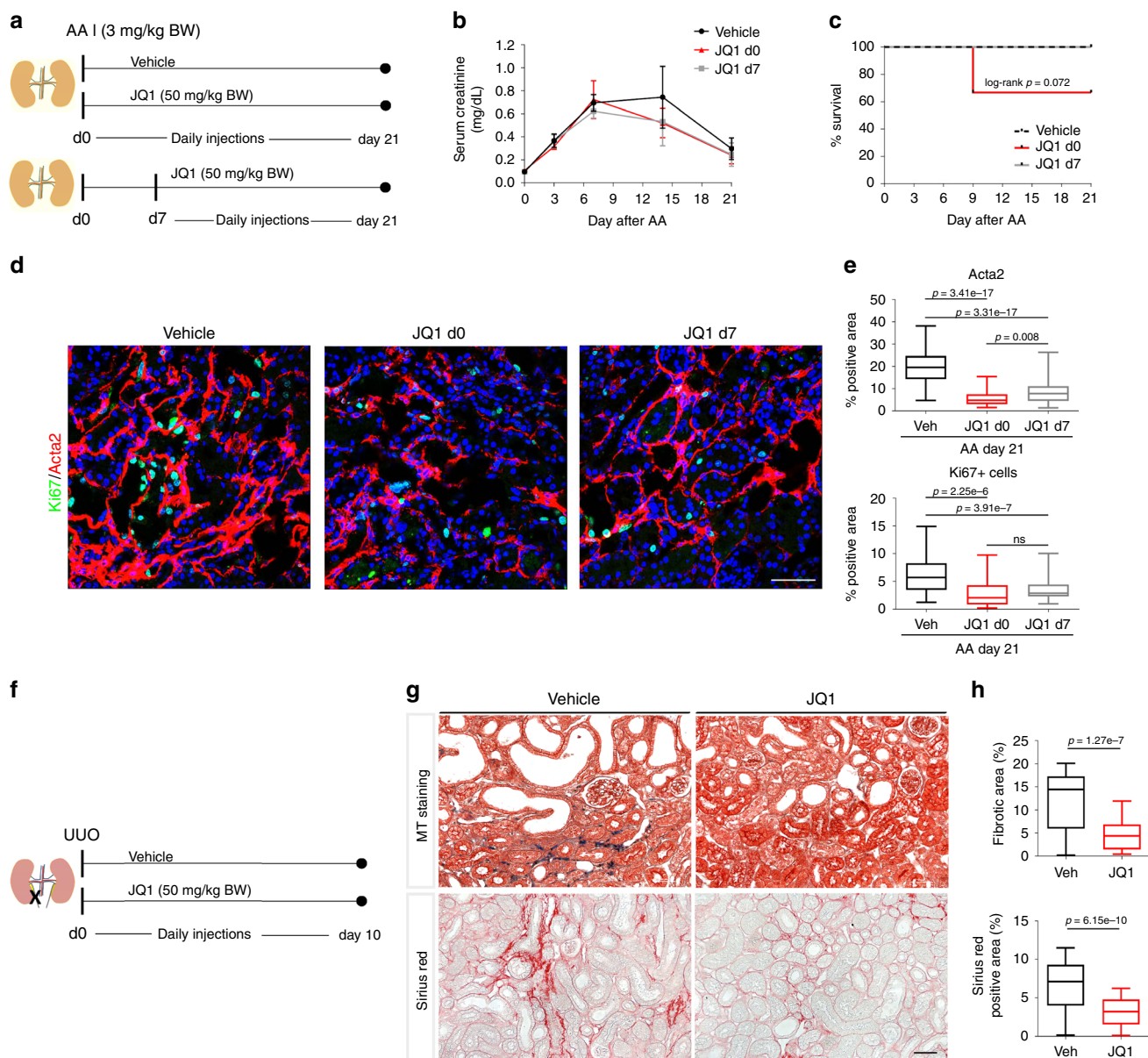

**Fig. 7 BET inhibition blocks fibrosis development in AAN and UUO. a** BALB/c mice (10- to 12-week-old males) were treated daily with JQ1 (50 mg/kg) or vehicle (DMSO/10% ß-cyclo dextrin 1:10) starting on the day of aristolochic acid (AA) injection (3 mg/kg BW) or day 7 after AA injection. Mice were sacrificed on day 21 after AA injection. **b** Serum creatinine (mg/dL) trajectories of vehicle and JQ1 treated mice from day 0 until day 21 after AA (mean ± SD). vehicle: n = 7; JQ1 d0: n = 6; JQ1 d7: n = 8 biologically independent samples. **c** Survival curves after AA injection: 100% survial in vehicle and JQ1 d7 group, 67% survival in mice treated with JQ1 from day 0 until day 21 after AA injection. **d** Representative Ki67-1/Acta2-immunostained AAN kidneys treated with vehicle or JQ1 d0 or JQ1 d7. **e** Quantification of α-SMA+ (Acta2) surface area. Percentage of Ki67+ cells (Ki67+ cells/total number of cells (DAPI) per hpf). vehicle: n = 7 (60 hpf); JQ1 d0: n = 4 (33 hpf); JQ1 d7: n = 8 (64 hpf) at least 8 hpf per sample. **f** C57BL/6N mice (8- to 10-week-old males) were treated daily starting at the day of UUO surgery with JQ1 (50 mg/kg) or vehicle, and were sacrificed on day 10 after surgery. **g** Representative trichrome-stained and Sirius-red stained sections. **h** Quantification of fibrotic area (masson trichrome +-stained) (n = 7, 5 hpf per sample) and Sirius red+ area (collagen) (n = 7, at least 7 hpf per sample. t-test (two-sided). Data represent the mean ± min, max. Box contains 50% of the data. Scale bars: 100 μm (**d**), 50 μm (**g**). Source data are provided as a Source data file.

binding next to Junb, a member of the AP-1 transcription factor family. AP-1 transcription factor motifs were highly enriched at injury-induced enhancer elements. Further, AP-1 transcription factors can complex with the GR leading to either activation or repression of target genes[50]. GR binding is highly dynamic at lost and gained enhancer elements after injury suggesting a crosstalk between GR and AP-1 transcription factors during kidney repair. We provide data of the binding of these transcription factors at enhancer sites in the kidney and novel clues as to how these

regulatory circuits orchestrate transcriptional kidney programs and response to injury.

Emerging evidence indicates that enhancer and super-enhancer regulation is crucial in health and disease[31] and directs cell-specific gene expression[30]. Here we show that enhancer and super-enhancer formation is BRD4 dependent in the kidney before and after kidney injury. Importantly, BET inhibitor administration had very different molecular effects in the kidney depending on timing after injury. BET inhibition in the early

recovery phase after AKI leads to impaired recovery and increased mortality whereas later after injury BET inhibition can block development of fibrosis. These different effects may be explained through the modification of the chromatin state of the cell by BET inhibitors[43,51]. It is interesting that KIM-1 expression is reduced with JQ1 treatment since early upregulation of KIM-1 after injury is protective in IRI[28]. Liu et al. have suggested a protective effect of JQ1 treatment administered seven days before IRI since they measured a decrease in creatinine level rise 24 h after IRI[52]. However, in their study survival after IRI was not assessed beyond 24 h after injury. Furthermore, pretreatment with JQ1 for 7 days may change the chromatin state of the renal cells and thereby alter the response to ischemia.

Studies in humans have documented broad changes in plasma protein concentrations 24 h after abapetalone (BET inhibitor) administration in CKD patients and healthy controls[53]. Osteopontin (Spp1) was one of the most affected proteins and was significantly down-regulated with the BET inhibitor in CKD and control patients, suggesting super-enhancer dependency of Spp1 and sensitivity to BET inhibitors also in humans. In general this study and various pre-clinical studies published by us and others present a rationale for antifibrotic effects of BET inhibitors and therefore a rationale for further human clinical investigation of this class of agents in CKD[45–47,54,55]. Different BET inhibitors, which specifically inhibit either the first (BD1) or the second (BD2) bromodomain of the BET proteins can have different functional contributions to the biological effects of BET inhibitors[56]. BD2 inhibitors were predominantly effective in inflammation-related disorders[56]. Apabetalone is an example of a BD2 inhibitor used in clinical trials[57]. Further, different genomic BRD4 binding and loads of cell-state-determining enhancers, depending on their molecular context in the kidney cell, can dramatically affect the outcome of the BET inhibitor therapy.

For patients at risk for AKI, our data call attention to potential caveats for use of small molecule inhibitors of BET proteins that are currently being tested in clinical trials, primarily in cancer. As AKI is quite common among hospitalized and cancer patients this requires further considerations and might effect clinical management.

In summary, we describe enhancer and super-enhancer elements and transcription factor binding at enhancer elements in the kidney before and after injury in vivo, yielding important clues to the regulation of cell-fate of kidney cells after injury. This work opens up a level of transcriptional understanding of kidney repair. In addition, the information obtained provides targets for therapeutic intervention.

## Methods

**Animal experiments.** Bilateral IRI was induced in male C57BL/6N mice by clamping of both renal arteries for 26 min at 37 °C with a retroperitoneal approach using pentobarbitol anesthesia. Reperfusion was verified by visual inspection of kidneys. One ml of 37 °C saline was administered by intraperitoneal injection (IP) after surgery for volume repletion. In SHAM operations both kidneys were exposed without induction of ischemia. For the unilateral ureteral obstruction model the left ureter of male C57BL/6N mice was exposed and ligated with a non-absorbable suture. One ml of 37 °C saline was given (IP) after surgery for volume repletion. Aristolochic Acid nephropathy was induced by a single injection of 3 mg/kg aristolochic acid I in saline intraperitoneally in male BALB/c mice. The BET inhibitor JQ1 or vehicle was provided to us by Dr. Jay Bradner (formerly at Brigham and Women's Hospital now at the Novartis Institutes for Biomedical Research). JQ1 was administered daily during the treatment periods. Pharmacokinetic studies in mice have shown that a dosage of 50 mg/kg/day of JQ1 is sufficient to achieve an effective blockade of BRD4[20,31]. We prepared a stock solution of JQ1 (50 mg/ml in DMSO) and further diluted it in 10% ß-cyclo-dextrin 1:9 to achieve solubility of the compound. 6 to 12 mice were used in each group in most experiments. Male C57BL/6 N or BALB/c mice aged 8 to 10 weeks weighing 20–22 g were purchased from Charles River Laboratories. Mice were kept under standard housing conditions (22–24 °C ambient temperature, 50-60% humidity, 12 h dark/light cycle). All mouse work was performed in accordance with all relevant ethical regulations for animal testing and research. The animal use

protocol was approved by the Institutional Animal Care and User Committee of the Harvard Medical School/Brigham and Women's Hospital.

**ChIP-seq.** Chip assays were performed as described previously[58]. Briefly, snap-frozen kidney cortex from SHAM or IRI mice (day 2) were ground into powder in liquid nitrogen. After 15 min chromatin cross-linking was carried out in 1% formaldehyde at room temperature and the reaction was quenched with 0.125 M glycine. After cell lysis and 22 min of sonication, sheared chromatin was incubated with 10 μg of following antibodies: anti-H3K27ac (Abcam, ab4729), anti-H3K4me3 (Millipore, 17-614), anti-BRD4 (Bethyl Laboratory, A301-985A), anti-RNA Polymerase II (Abcam, ab5408), anti-BRD2 (Bethyl Laboratory, A301-583A), anti-BRD3 (Bethyl Laboratory, A301-368A), anti-HNF4A (Abcam, ab41898), anti-GR (Thermo Scientific, PA1-511A), anti-STAT3 (Santa Cruz, sc-482) and anti-STAT5 (Santa Cruz, sc-835). Libraries for next-generation sequencing were prepared with NEBNext Ultra II DNA Library Prep Kit (New England BioLabs, E7645) and sequenced on HiSeq 2500 machine (Illumina).

**ChIP-seq data analysis.** The ChIP-seq analysis workflow comprised trimming using trimmomatic (version 0.36)[59] in order to filter low quality reads, followed by the application of bowtie aligner (version 1.1.2)[60] with the parameter of -m 1 to retrieve only uniquely mapped reads (mm10 reference genome), and finally Homer software (version 4.8.2)[61] and Integrative Genomics Viewer (version 2.3.81)[62] for the visualization. DeepTools (version 2.1.0)[63] was used to obtain the correlation between the replicates.

In order to identify regions of ChIP-seq enrichment over the background, the MACS2 (version 2.1.1)[64] peak finding algorithm was used. Broad peak calling of H3K27ac with a q-value cutoff of 0.1 and 0.05 for SHAM and IRI day 2 was done in replicates, which were subsequently overlapped using BEDtools (version 2.26.0)[65] to identify high-confident peaks. Promoters were filtered using the coordinates of the GFF file and overlapping them with the peak files using BEDtools. Annotation of the identified enhancer elements to their target genes was done with GREAT[27] using the setting of "Basal plus extension" and a maximum distal distance of 100 kb. Coverage plots (normalized to 10 million reads) and motif analysis with default background were done using Homer software (version 4.8.2)[61]. HOMER[61] was used for motif analysis using the default settings.

The ROSE algorithm[30,31] was applied for super-enhancer analysis. We used enhancer elements identified by H3K27ac, and the default stitching size of 12.5 kb and H3K27ac and BRD4 BAM files as input. Graphs were generated using R (R Project for Statistical Computing, https://www.R-project.org/) and the packages ggplot2[66] and dplyr (https://CRAN.R-project.org/package=dplyr). Box plots were used to illustrate data (Median, middle bar inside each box; IQR (interquartile range)), the box containing 50% of the data; whiskers, 1.5 times the IQR). Two sample t-tests was applied where appropriate. One-way ANOVA with Tukey post hoc test was applied where appropriate.

**Gene expression profiling.** Total RNA was isolated from kidney cortex samples with the PureLink RNA Mini Kit (Thermo Fisher Scientific, 12183018A) or TRIzol reagent (Thermo Fisher Scientific, 15596029) for RNA-Seq or qRT-PCR. For qRT-PCR 2 μg of total RNA was reverse transcribed with the M-MLV reverse transcriptase Kit and oligo dT primers (Invitrogen). Gene expression was analyzed by qPCR using SYBR Green reagent (Bio-Rad) with gene-specific primers in 25 μl reactions in triplicate on Bio-Rad real-time detection system.

Libraries for next-generation sequencing were prepared with TruSeq Stranded RNA LT Kit (Illumina, 15032611) and sequenced on HiSeq 2500 machine (Illumina).

**RNA-seq analysis.** RNA-seq data were trimmed using trimmomatic (version 0.36)[59] and subsequently aligned to the reference genome mm10 applying STAR RNA-seq aligner (version 2.5.3a)[59]. HTSeq (version 0.6.1p1)[67] and DESeq2[68] were applied for the RNA-seq analysis.

**Renal function and histology.** The plasma creatinine and BUN concentration was determined by Mass Spectrometry at the University of Alabama and BUN infinity kit from ThermoFisher Scientific, respectively. Kidney histology was examined on methacarn-fixed sections stained for PAS, H&E, Masson's trichrome and Sirius red. The degree of interstitial fibrosis was scored quantitatively on Masson's trichrome stained tissue or Sirius red-stained tissue (Collagen) with Image J.

**Immunofluorescence staining.** Immunofluorescence staining of the kidney was performed on fixed-frozen sections. Briefly, the tissue sections were activated and labeled with antibodies, including rabbit anti-Ki-67 (Vector, VP-K451, 1 in 200), rabbit anti-KIM-1 (R9[25], 1 in 200), rabbit anti-α-SMA (Sigma, 1 in 400), rat anti-F4/80 (Abcam, ab6640, 1 in 1000), rabbit anti-Hnf1b (Thermo Fisher Scientific, 720259), rabbit anti-GR (Thermo Fisher Scientific, PA1-511A), rabbit anti-Slc34a1 (Novusbio, NBP2-13328), rabbit anti-Spp1 (Abcam, ab8448) and rat anti-Kl (BioLogo, KM2076). The slides were then exposed to FITC or Cy3-labeled secondary antibodies (Jackson ImmunoResearch). The staining was examined with fluorescence microscopes (Nikon TE 1000 and Nikon C1 confocal). At least 7

high-power fields/section for each sample were examined in each evaluation and quantification (positive area or cell count) was performed with an in-house Macro in Image J. Spp1 plasma concentration was measured by ELISA (R&D Systems, MOST00).

**Reporting summary**. Further information on research design is available in the Nature Research Life Sciences Reporting Summary linked to this article.

## Data availability

The data that support this work is available from the corresponding authors upon reasonable request. ChIP-seq and RNA-seq raw data files are available in the Gene Expression Omnibus (GEO) at NCBI with the accession number GSE114294. The source data underlying Fig. 5b, e, 6b, g, h, 7b, e, h and Supplementary Figs. 3b, c and 8b–d are provided as a Source data file.

## Code availability

A summary of our analysis approach can be found in the Supplementary information. We only applied existing tools to analyze our data.

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

## Acknowledgements

This research was supported by a Marie Curie International Outgoing Fellowship within the 7th European Community Framework Programme (#328613) and the Austrian Science Fund (FWF) P30373 to J.W., US National Institutes of Health awards DK039773, DK072381, and TR002155 to J.V.B. and through the Intramural Research Programs of NIDDK/NIH (MW, HKL, LH). This work utilized the computational resources of the NIH HPC Biowulf cluster. (http://hpc.nih.gov).

## Author contributions

J.W., L.H., and J.V.B. designed the experiments, and J.W. carried out the majority of the experiments. M.W. and HK.L. performed the computational analyses, and H.O., J.J., and T.I. performed immunofluorescence staining of some of the samples. M.T.V. contributed important insights and helped with data interpretation. R.E. reviewed the paper. J.W., M.W., and J.V.B. wrote the paper. All authors carefully reviewed the manuscript.

## Competing interests

J.V.B. is co-inventor on KIM-1 patents assigned to Partners Healthcare. He is co-founder of Goldfinch Bio. He is a consultant for Cadent, Aldeyra and an advisor with equity in Medibeacon Inc, Rubius, Theravance, Goldilocks, DxNow, and Sentien. All other authors have no competing interests.
