## [Peer Review File · Nature Communications]

Reviewers' comments:

Reviewer #1 (Remarks to the Author):

This manuscript by Wilflingseder et al primarily addresses the transcriptional responses to acute kidney injury following a period of ischaemia. The main focus of the manuscript is understanding the gene expression changes seen with reperfusion injury (IRI). The conclusions drawn are that (i) there are major gene expression changes following IRI which are associated with the appearance of new cis-regulatory elements and (ii) BRD4 plays a role in facilitating the transcriptional responses to IRI. Interestingly, BET inhibition exacerbates the AKI following ischaemia leading to increased mortality in the animals, leading the authors to suggest that the BET proteins are critical to the recovery from IRI. The later part of the manuscript looks at the consequences of fibrosis induced by ureteric obstruction (UUO) and delivery of a nephrotoxin (AAN) and here they show that the BET proteins likely facilitate fibrosis because BET inhibition ameliorates the fibrotic process.

The strength of the manuscript is that all of the experiments are done in primary tissues from mice with experimental induction of reperfusion injury and uretic obstruction. However, the significant weakness of the manuscript is that it is entirely correlative. The substantial portion of this manuscript (first 4 figures) is a standard analysis of 4 ChIP-seq experiments and some RNA-seq data. Whilst there is no doubt that there are major transcriptional responses to ischaemia, in part this is not surprising as this would be expected of any injurious process which is accompanied by repair. What is left unexplored is exactly what genes are the major drivers of this process and how the BET proteins regulate the expression of these genes. By this I mean, the various injuries induced IRI, UUO and aristocholic acid nephrotoxicity (AAN) probably cause different transcriptional changes, the response to these injuries are likely mediated by different pathways and gene expression changes - the precise role that BET proteins play beyond being detectable by ChIP at neighbouring enhancers to some of these genes is critical but currently unexplored.

The BET proteins are major transcriptional regulators and any process that requires an acute transcriptional response is likely to require the BET proteins (and other general transcriptional coactivators). This is why BET inhibitors have been shown to ameliorate a vast array of pathologies ranging from myocardial ischaemia to sepsis. It maybe that targeting any relatively generic transcriptional regulator does the same thing. For instance, CDK9 or CDK7 inhibitors would be good controls to see if the results shown here are specific to BET proteins or just reflect the fact that anything that negates the normal physiological transcription response to injury would do the same thing. In the absence of at least some mechanistic insight that helps the readers make some conclusions on what specific genes (not hundreds of computationally predicted SE) mediate the response to the different injuries and whether the BET proteins play a specific role in this process the main message of the paper is pretty thin in its current form.

Specific points:

The majority of the first 4 figures could be condensed into 2 main figures at most and the rest put into Supplementary data. For instance, Fig.1e,f is not helpful to the reader.

Figure 2 is largely unhelpful – although there are many computationally predicted TF none of these are actually explored in a functional experiment. i.e what happens to the IRI response if the injurious process was done in animals that are genetically null for FRA1 or any of the other AP1 members shown? Although genes such as Lcn2, Havcr1, Spp1 etc.. are highlighted in the manuscript, their role in this process is not explored.

What are the transcriptional changes seen after UUO and AA treatment? How do they differ from IRI? What about BRD2 and BRD3 both of which have been shown to play critical roles in mediating inflammatory responses what roles do they have in IRI, UUO and AAN? Do the predicted transcription factors from the 'new superenhancers' actually bind these SE? Do they recruit the

BET proteins?

Even if some of these questions are addressed it would elevate the value of the manuscript to the field.

Reviewer #2 (Remarks to the Author):

In this manuscript, the authors characterize the dynamics of enhancers and super enhancers during the repair phase following ischemic acute kidney injury (AKI) in mice. Overall, this is a very comprehensive study, very thorough and well designed. Also, it is a novel approach to the understanding of the pathogenesis of renal injury and repair which I am sure it will be very much appreciated by the AKI community.

The manuscript is divided into two major areas, the first aims to characterize the dynamics of enhancer and super enhancer areas in gene regulation during the repair phase and the second part, a little more interventionist aimed to determine the specific role of BRD4 in the repair process.

While I have little or no concerns regarding the first part of the manuscript (again, a very thorough design and methodology), I have some issues on how the authors try to characterize the role of BRD4. More specifically, authors employ JQ1, a general BET inhibitor, before and 1 day after ischemic AKI and find that it impairs proper recovery of the kidney accelerating death following AKI. Based on this, authors assume that the effects of JQ1 exacerbating AKI are secondary to the blockade of BRD4. However there are two major conflicting issues with this finding:

1) Authors provide no evidence of BRD4 target engagement with JQ1, based on the authors hypothesis, it would be very relevant to determine whether JQ1 affect the enhancer dynamics associated with BRD4. For example, does JQ1 reduce the coverage of BRD4 in the promoter region of *Havcr1* and *Spp1* shown in figure 3i and 4e? Similarly, is *Spp1* expression down-regulated in JQ1 treated mice? Is BRD4 expression in renal epithelia altered in AKI and is it changed by JQ1 treatment? If no proper demonstration that BRD4 is inhibited by JQ1 in AKI could suggest that the deleterious effects could be due to off-target nephrotoxic effects of JQ1 which are further exacerbated in mice undergoing renal ischemia.

2) Show efficient target engagement with JQ1 is also important in this study as the data provided is just the opposite from a recent paper from Liu et al (<https://www.sciencedirect.com/science/article/pii/S2213231719300618#fig1>) in which they demonstrate marked protective effects of JQ1 in iAKI. In that paper, authors show that BRD4 is induced in ischemic AKI and a dose effect of JQ1 in BRD4 expression and a correlation between JQ1 and renal injury. A proper discussion in the disparity of the results should be included to the manuscript as well. I understand that the authors in the Liu paper have a longer pretreatment phase (7 days) before ischemic insult which may substantially alter the dynamics of BRD4.

I think that if authors are able to provide evidence of proper target engagement with JQ1 in vivo, it should be sufficient for the community to realize that the effects of JQ1 observed in this study are truly secondary to BRD4 blockade and not just due to the intrinsic toxicity of this compound.

Minor concerns;

1) Changes in gene expression while of interest should be correlated with changes in protein expression. This is important as at day 2 post IR some proteins with long half lives like membrane proteins (*slc34a1* for example) may be still present and active even after substantial down-regulation of their mRNA. Authors should provide evidence for protein expression for the main genes of interest provided in this manuscript (*slc34a1*, *spp1*, *havcr1*, *KI*) to ascertain that the changes in gene expression are actually translated into changes in protein levels at day 2 during

repair.

2) The title should indicate ischemic acute kidney injury and not acute kidney injury, as authors indicate In the text, there are multiple causes of AKI and in this manuscript authors only focus on ischemia, perhaps other inducers show different dynamics than the ones observed in this manuscript

3) Please define BRD4 in the abstract and FDR in the text

4) The UUO data with JQ1 is definitely not novel and should identify in the text that the data is similar to that from previous studies as in references 62 and 63 of the manuscript. These references already show previous evidence of JQ1 protective effects in fibrosis in the UUO model, so figure 7A is not providing more data beyond what is already published.

Reviewer #3 (Remarks to the Author):

In order to determine transcriptional regulators of the repair response after acute kidney injury, the authors of this manuscript profiled enhancer and super enhancer repertoire in uninjured and repairing kidneys on day two following ischemia reperfusion injury. They identified several transcription factor binding sites on enhancer and super enhancer regions. In order to establish role of BRD4 on the super enhancer function authors have used BET inhibitor JQ1. They show that JQ1 (unexpectedly) increased mortality at day two and three after ischemia reperfusion injury compared to pre ischemia reperfusion injury. On the other hand, the found JQ1 can block kidney fibrosis in unilateral ureter obstruction and aristolochic acid kidney injury models.

Altogether authors nicely showed the role of BET proteins in this AKI study using a relevant mice model system. Overall the manuscript is well written and conclusions have translational significance especially for needed caution while evaluating future BET inhibitor therapies for AKI.

The in vivo studies are good. On the other hand, authors need to do a more detailed characterization of the enhancers and super-enhancers in their model system, esp as this is the more novel part. Moreover more experiments are need to demonstrate the connection between the enhancer profiling and the JQ1 interventions. Specific comments, suggestions and points to be addressed to further strengthen their manuscript are detailed below.

1. Authors need to experimentally validate some of the differential enhancers in SHARED, IRI decreased and IRI increased groups by performing ChIP with H3K27ac, BRD4 and Pol II antibody at candidate target and control regions.

2. There should be some experimental connection between the first part of the paper and second part (JQ1 treatment). The authors should show whether some of the critical enhancers and superenhancers that are altered in the AKI models and differential expression their putative associated genes are differentially affected by JQ1 treatment (using samples from the JQ treated mice). Comparison of such factors during early IRI (inflammation) and in the fibrosis models could give valuable clues underlying the differential response to JQ1 and the associated genes in the AKI versus fibrosis models. This in turn could potentially help design better BET inhibitors for AKI.

3. Figure 2: i.e. Identification of the transcription factor on the enhancers. This Figure can be part of some other main/supplemental figure. Although the data is interesting, it does not need an independent main figure. Authors need to perform ChIP/ChIP-seq with SHARED (HNF4a) and IRI decreased and increased (any bZIP family) transcription factors and show their specific binding on the enhancer regions. Are these transcription factors differentially regulated in SHAM and IRI treated kidney samples? Are the identified transcription factors specific for enhancer binding or they are also enriched in promoters of differentially regulated genes?

4. Many or all the important enhancer-associated genes should to be validated experimentally

(example Figure 3F).

5. It is important to show that these enhancers are functional. I suggest performing luciferase assays by using the enhancer constructs (For example 3g, 3h and 3I).

6. Authors can mutate or delete the super-enhancer or constituent enhancer regions of SPP1 and HNF1b and show these super-enhancers are specific to their associated genes.

7. Since JQ1 can affect other BET proteins besides Brd4, Did authors check the expression of candidate BET proteins in SHAM and IRI treated kidney samples?

8. Can authors comment on the role of other BET proteins while demonstrating JQ1 role in transcriptional kidney repair programs?

9. In the methods section, please provide sonication conditions and antibody concentrations used for ChIP.

Reviewers' comments:

Reviewer #1 (Remarks to the Author):

This manuscript by Wilflingseder et al primarily addresses the transcriptional responses to acute kidney injury following a period of ischaemia. The main focus of the manuscript is understanding the gene expression changes seen with reperfusion injury (IRI). The conclusions drawn are that (i) there are major gene expression changes following IRI which are associated with the appearance of new cis-regulatory elements and (ii) BRD4 plays a role in facilitating the transcriptional responses to IRI. Interestingly, BET inhibition exacerbates the AKI following ischaemia leading to increased mortality in the animals, leading the authors to suggest that the BET proteins are critical to the recovery from IRI. The later part of the manuscript looks at the consequences of fibrosis induced by ureteric obstruction (UUO) and delivery of a nephrotoxin (AAN) and here they show that the BET proteins likely facilitate fibrosis because BET inhibition ameliorates the fibrotic process.

The strength of the manuscript is that all of the experiments are done in primary tissues from mice with experimental induction of reperfusion injury and uretic obstruction. However, the significant weakness of the manuscript is that it is entirely correlative. The substantial portion of this manuscript (first 4 figures) is a standard analysis of 4 ChIP-seq experiments and some RNA-seq data. Whilst there is no doubt that there are major transcriptional responses to ischaemia, in part this is not surprising as this would be expected of any injurious process which is accompanied by repair. What is left unexplored is exactly what genes are the major drivers of this process and how the BET proteins regulate the expression of these genes. By this I mean, the various injuries induced IRI, UUO and aristocholic acid nephrotoxicity (AAN) probably cause different transcriptional changes, the response to these injuries are likely mediated by different pathways and gene expression changes - the precise role that BET proteins play beyond being detectable by ChIP at neighbouring enhancers to some of these genes is critical but currently unexplored.

The BET proteins are major transcriptional regulators and any process that requires an acute transcriptional response is likely to require the BET proteins (and other general transcriptional coactivators). This is why BET inhibitors have been shown to ameliorate a vast array of pathologies ranging from myocardial ischaemia to sepsis. It maybe that targeting any relatively generic transcriptional regulator does the same thing. For instance, CDK9 or CDK7 inhibitors would be good controls to see if the results shown here are specific to BET proteins or just reflect the fact that anything that negates the normal physiological transcription response to injury would do the same thing. In the absence of at least some mechanistic insight that helps the readers make some conclusions on what specific genes (not hundreds of computationally predicted SE) mediate the response to the different injuries and whether the BET proteins play a specific role in this process the main message of the paper is pretty thin in its current form.

We thank the reviewer for their kind remarks concerning our study.

While it is true that it is not surprising to see a major transcriptional response to acute kidney injury, our main goal was to identify enhancer and super-enhancer elements during this process, which is novel for the characterization of the kidney response to injury. We very much appreciate that the injury and repair process is complex and yet the data with BET inhibition indicates that the processes that we are affecting by this inhibition have key adaptive roles in early repair after injury and contributions to the fibrosis that results from maladaptive repair. Our experiments suggest an important role of super enhancers in the well-appreciated clinical transition of acute kidney injury (AKI) to fibrotic chronic kidney disease (CKD). Thus we were not only interested to characterize the enhancer elements, but also to provide proof of principle for the role of the identified enhancer elements in the repair process on a genome-wide scale. For this purpose we used BRD4 as marker for enhancer elements and the BET inhibitor JQ1 to elucidate the role of the identified enhancer dynamics in the repair process. Hence we agree with the Reviewer that further insights into the regulation of key repair pathways would add value to the current manuscript.

To respond to the Reviewer's important comments we added several additional novel components to the manuscript by reporting the presence of predicted transcription factors at the identified enhancer elements and provide data on the dynamics of changes in binding of them after injury. We used our in silico motif analysis to choose transcription factors for further ChIP-seq analysis, namely hepatocyte nuclear factor 4 alpha (HNF4A), glucocorticoid receptor (GR), and signal transducer and activator of transcription (STAT) 3 and 5.

Furthermore, as suggested by the Reviewer, we tested CDK9 inhibition with flavopiridol (2.5 mg/kg and 1 mg/kg body weight per day) in our IRI model. It was shown recently that BET proteins act as master transcription elongation factors independent of CDK9 recruitment¹. Unlike the mortality results presented with JQ1 we observed no significant increase in mortality with two tested concentrations of flavopiridol when we started the treatment 3 hours before the surgery. Therefore this shows that there is some level of specificity to the BET inhibitor effect since another agent (CDK9 inhibitor), which inhibits transcriptional responses have a significantly different effects on animal survival. We included this important control, suggested by the Reviewer in the manuscript. Furthermore, the lack of mortality with JQ1 administered more remotely from the initial ischemia points to the lack of generalized toxicity and further points to the inhibition of early repair as an explanation for the mortality seen with early exposure after injury. We found that KIM-1 expression is downregulated by early treatment with JQ1 (Figure 6g,h) and our lab has previously published that KIM-1 expression is reduced with JQ1 treatment since early upregulation of KIM-1 after injury is protective in IRI².

Specific points:

The majority of the first 4 figures could be condensed into 2 main figures at most and the rest put into Supplementary data. For instance, Fig.1e,f is not helpful to the reader.

As suggested we condensed the first figures in the manuscript. Figure panel 1e and f, showing the correctness of the enhancer classification in the three groups (SHARED, IRI decreased and IRI increased), are now in the supplemental material.

Figure 2 is largely unhelpful – although there are many computationally predicted TF none of these are actually explored in a functional experiment. i.e what happens to the IRI response if the injurious process was done in animals that are genetically null for FRA1 or any of the other AP1 members shown? Although genes such as Lcn2, Havcr1, Spp1 etc.. are highlighted in the manuscript, their role in this process is not explored.

A major new addition to the manuscript is the analyses of transcription factor ChIP-seq data. Based on our computationally predicted transcription factor motifs (Suppl. Fig. S5 and Suppl. Table S9) and availability of suitable antibodies we chose a number of important transcription factors, for further ChIP-seq experiments, namely HNF4A, GR, FRA1, FOSL2, JUN, STAT3 and STAT5. We were able to generate high quality ChIP-seq for transcription factors which have important roles in kidney epithelial cell fate and response to injury - HNF4A³, GR⁴, STAT3⁵ and STAT5. Most important we can show binding of the predicted transcription factors at the identified enhancer elements and a dynamic occupancy after injury. HNF4A and GR binding is decreased at IRI decreased enhancer elements and STAT3 binding increases at IRI increased enhancer elements (Figure 4a). We cannot observe a dynamic binding after kidney injury in STAT5 (Figure 4a). Representative examples of transcription factor binding before and after injury are shown in Figure 4b and Suppl. Figure S6.

The antibodies for AP-1 transcription factors failed to be of suitable quality for ChIP-seq analysis. However, according to the literature some members of the AP-1 transcription factor family have been already evaluated in kidney injury and repair⁶⁻⁸. For example the JunD/Fos complex showed protective properties in acute and chronic kidney injury models^{6,8}.

Lcn2, Havcr1 and Spp1 are among the most up-regulated transcripts after kidney injury and have been evaluated as biomarkers of injury⁹⁻¹¹. Havcr1 and Spp1 have an adaptive role in the early phase of kidney repair^{2,12,13}. Our goal was to shed light on the transcriptional regulation of these transcripts by identifying important enhancer elements and transcription factors involved in regulation. We found that GR binding was present at the regulatory element of Havcr1 before injury and decreased after Havcr1 transcriptional activation after injury. At the super-enhancer of Spp1 we found dynamic binding of GR with decreased binding after injury and increased in STAT3 binding after injury (Suppl. Figure S6a).

Figure 4

Figure 4 – Genome-wide analyses of HNF4a, GR, STAT3 and STAT5 binding at enhancer elements before and after kidney injury. (a) Coverage plots at all, SHARED, IRI decreased and IRI increased enhancers. The coverage for HNF4a and GR decreased in SHARED and IRI decreased enhancer group after IRI. In contrast, STAT3 shows increased coverage at IRI increased enhancers. STAT5 peak height at enhancer elements is unchanged after kidney injury. **(b)** Representative examples of transcription factor binding at enhancer sites in kidney epithelia cells. The *Slc34a1* and *Junb* genomic locus are shown for HNF4A, GR, STAT3 and STAT5 binding together with H3K4me3 and H3K27ac in the SHAM (left) and IRI (right) condition. Enhancer elements are indicated by grey bars.

Suppl. Figure S6. Representative examples of transcription factor binding at super-enhancer sites in kidney epithelia cells. (a) The Kl, Havcr1, Spp1 and Hnf1b genomic locus are shown for HNF4A, GR, STAT3 and STAT5 binding together with H3K4me3 and H3K27ac in the SHAM (left) and IRI (right) condition. **(b)** Representative control regions (Bcl6 and Neat1) are shown for transcription factor ChIP-seq quality between SHAM and IRI. Enhancer elements are indicated by grey bars.

What are the transcriptional changes seen after UUO and AA treatment? How do they differ from IRI?

While it is true that IRI, UUO and AAN represent different proximate causes of kidney injury they share significant transcriptional changes underlying overlapping and converging mechanisms of kidney injury and repair¹⁴⁻¹⁶. The time points when certain transcriptional changes occur can be different among the models reflecting the differences in cause, severity and level of reversibility of the injury¹⁵⁻¹⁷. Transcriptional profiles of acute kidney injury models are intensively studied and are publicly available at Gene Expression Omnibus or ArrayExpress with (in total) more than 200 datasets. Response to hypoxia in kidney epithelia cells is part of the shared transcriptional program although hypoxia only models one aspect of IRI. Further indication that response to kidney injury overlaps among the models is the shared increase of kidney injury biomarkers such as KIM-1 (Havcr1), decreased expression of Klotho and increased number of proliferating cells (Ki67+) in all models. In addition the long-term consequences of kidney injury and repair are the development of interstitial fibrosis and tubular atrophy representing common pathways converging as acute kidney injury morphs into chronic kidney disease in all models^{15,18,19} as it does in human kidney disease²⁰. Despite the shared transcriptional programs there are differences in the onset and severity of kidney fibrosis which was one of the reasons to include the AAN and UUO model in the evaluation of JQ1 on the development of kidney fibrosis in our study.

What about BRD2 and BRD3 both of which have been shown to play critical roles in mediating inflammatory responses what roles do they have in IRI, UUO and AAN? Do the predicted transcription factors from the 'new superenhancers' actually bind these SE? Do they recruit the BET proteins?

To elucidate the importance of the different BET family members (BRD2, BRD3 and BRD4) in the kidney we performed additional immunofluorescence staining and ChIP-seq experiments. We found that BRD4 is the dominant member of the BET family in the kidney with higher protein abundance and genome-wide binding at regulatory elements compared to BRD2 and BRD3 which have low protein abundance and low binding at the genome before and after kidney injury (IRI). Please see Suppl. Figure S7

We could show that selected predicted transcription factors can bind at the identified enhancer and super-enhancer elements. Further binding after injury of the transcription factors HNF4A, GR and STAT3 is dynamic. ChIP-seq profiles for H3K27ac, H3K4me3, BRD4, BRD3, BRD2, POL 2, HNF4a, GR, STAT3 and STAT5 in replicate in SHAM and IRI day 2 (40 profiles) can be

downloaded at Gene Expression Omnibus GSE114294, which allows you to look up any gene or genomic region of interest.

The recruitment hierarchy at enhancer or promoter sites is subject of ongoing research and has not been established. The most pressing question is which must be first: open chromatin or transcription factor binding ²¹. The recruitment of BET proteins likely occurs after chromatin activation and transcription factor binding as part of the transcription initiation complex. However this question is out of scope of the current study.

Suppl. Figure S7. Assessment of BET family members: BRD4, BRD2 and BRD3. (a) Representative immunostaining of BRD4, BRD2 and BRD3 in kidney cortex in SHAM and IRI samples at day 2 after injury (Scale bar: 50 μ m) and (b) Genome-wide coverage blots of BRD4, BRD2 and BRD3 ChIP-seq profiles in SHAM and IRI. BRD4 is the dominant member of the BET family in kidney cortex.

Even if some of these questions are addressed it would elevate the value of the manuscript to the field.

We thank the Reviewer for pointing out the value of our study and for offering many helpful suggestions.

Reviewer #2 (Remarks to the Author):

In this manuscript, the authors characterize the dynamics of enhancers and super enhancers during the repair phase following ischemic acute kidney injury (AKI) in mice. Overall, this is a very comprehensive study, very thorough and well designed. Also, it is a novel approach to the understanding of the pathogenesis of renal injury and repair which I am sure it will be very much appreciated by the AKI community.

The manuscript is divided into two major areas, the first aims to characterize the dynamics of enhancer and super enhancer areas in gene regulation during the repair phase and the second part, a little more interventionist aimed to determine the specific role of BRD4 in the repair process.

While I have little or no concerns regarding the first part of the manuscript (again, a very thorough design and methodology).

We appreciate the Reviewer's comments regarding the value of our delineation of the dynamics of enhancers and super-enhancers during repair after ischemic injury resulting in AKI.

I have some issues on how the authors try to characterize the role of BRD4. More specifically, authors employ JQ1, a general BET inhibitor, before and 1 day after ischemic AKI and find that it impairs proper recovery of the kidney accelerating death following AKI. Based on this, authors assume that the effects of JQ1 exacerbating AKI are secondary to the blockade of BRD4. However there are two major conflicting issues with this finding:

1) Authors provide no evidence of BRD4 target engagement with JQ1, based on the authors hypothesis, it would be very relevant to determine whether JQ1 affect the enhancer dynamics associated with BRD4. For example, does JQ1 reduce the coverage of BRD4 in the promoter region of *Havcr1* and *Spp1* shown in figure 3i and 4e? Similarly, is *Spp1* expression down-regulated in JQ1 treated mice? Is BRD4 expression in renal epithelia altered in AKI and is it changed by JQ1 treatment? If no proper demonstration that BRD4 is inhibited by JQ1 in AKI could suggest that the deleterious effects could be due to off-target nephrotoxic effects of JQ1 which are further exacerbated in mice undergoing renal ischemia.

The cell-permeable small molecule, JQ1, binds competitively with high potency and specificity to the acetyl-lysine recognition bromodomain motifs of the BET family (BRD4, BRD3, BRD2 and BRDT)²². BRDT is testis specific. The action and specificity of JQ1 was intensively studied in *in-vitro* and *in-vivo* models and shows effects in numerous phenotypes²³⁻³⁴. The effects are due to

the intrinsic mode of action of this small molecule, which targets BET proteins (chromatin readers) on a genome-wide scale. Therefore, the range of effected genes and phenotypes can be explained through the inhibition of the chromatin state of the cell rather than a specific pathway, ligand, kinase or receptor³⁵. Further, the mechanism of how JQ1 interferes with the transcriptional process has been reported³⁶. Through competitive binding of the two extra terminal bromodomains of the BET proteins JQ1 inhibits the transcriptional elongation process resulting in genome-wide Pol II pausing³⁶.

In response to the Reviewer's suggestions we studied the mode of action of JQ1 both directly on gene expression level with RNA-seq and additionally through Pol II binding on the gene body as measurement of the transcriptional elongation process, the functional consequence of BET inhibition. We could show that JQ1 leads to genome-wide Pol II pausing two days after IRI. As an example we show the *Spp1* gene body. Pol II has less coverage under JQ1 treatment. Furthermore *Spp1* is down-regulated by JQ1. Please see figure below and Figure 6i-k.

Figure 6i-k. Genome-wide assessment of Pol II binding. (i) Genome-wide coverage blots of Pol II on the gene body. Pol II binding after JQ1 treatment is increased at the TSS and decreased across the gene body indicating Pol II pausing **(j)** Pol II ChIP-seq tracks at the *Spp1* gene body **(k)** Fold change of *Spp1* after IRI comparing vehicle and JQ1 treated animals at day 2 after injury. *** P < 0.001

[2\) Show efficient target engagement with JQ1 is also important in this study as the data provided is just the opposite from a recent paper from Liu et al \(https://www.sciencedirect.com/science/article/pii/S2213231719300618#fig1\) in which they demonstrate marked protective effects of JQ1 in iAKI. In that paper, authors show that BRD4 is induced in ischemic AKI and a dose effect of JQ1 in BRD4 expression and a correlation between JQ1 and renal injury. A proper discussion in the disparity of the results should be included to the manuscript as well. I understand that the authors in the Liu paper have a longer pretreatment phase \(7 days\) before ischemic insult which may substantially alter the dynamics of BRD4.](https://www.sciencedirect.com/science/article/pii/S2213231719300618#fig1)

We thank the reviewer for this comment.

The study of Liu et al. has several differences when compared to our study. First the latest time point was 24 hours after injury which does not allow for the assessment of the survival rate after AKI. We did not see mortality until day 3. Further we cannot detect a significant increase in BRD4 gene expression in the kidneys after injury ($p = 0.09$). However, the binding pattern of BRD4 on the genome is much more important compared to mRNA or protein changes of BRD4. We detected a genome-wide change in the pattern of BRD4 measured by ChIP-seq after injury (Figure 1 and Figure 3). Further we detected no change in creatinine or BUN levels with JQ1 treatment whereas Liu et al. did see a difference. This might be due to the fact that the authors already started the treatment with JQ1 7 days before the injury and therefore already changed the chromatin state of the renal cells before the injury, as pointed out by the Reviewer. One can think of JQ1 treatment as a preconditioning event similar to previous ischemic injury. In a number of studies we have reported that ischemic preconditioning is protective against subsequent ischemic injury³⁷. We have included discussion of the Liu study in the Discussion section of the manuscript.

I think that if authors are able to provide evidence of proper target engagement with JQ1 in vivo, it should be sufficient for the community to realize that the effects of JQ1 observed in this study are truly secondary to BRD4 blockade and not just due to the intrinsic toxicity of this compound.

If the toxicity associated with early treatment after IRI were due to off-target toxicity they we would expect to see this when the drug is given at later times after IRI which we do not see. In fact when JQ1 is given more than 3 days after IRI there is protection against maladaptive fibrotic repair. Additional evidence that our observed phenotypic changes are on-target effects of BET inhibition are from genetic studies. After genetic silencing of BRD4 in mice a range of tissue homeostasis disruption can be observed³⁸. The on-target effects (toxicities) of genetic BRD4 silencing include epidermal hyperplasia, alopecia, and stem cell depletion in the small intestine. Interestingly mice with deletion of BRD4 are also more susceptible to stress (irradiation) and show reduced regeneration potential. Nephrotoxicity was not observed in these BRD4 silenced mice and in our SHAM JQ1 treated mice, as assessed by serum creatinine levels, histology or fibrosis development.

Minor concerns;

1) Changes in gene expression while of interest should be correlated with changes in protein expression. This is important as at day 2 post IR some proteins with long half lives like membrane proteins (slc34a1 for example) may be still present and active even after substantial down-regulation of their mRNA. Authors should provide evidence for protein expression for the main genes of interest provided in this manuscript (slc34a1, spp1, havcr1, KI) to ascertain that the changes in gene expression are actually translated into changes in protein levels at day 2 during repair.

We added results of additional experiments in the supplemental material showing decreased protein levels of KL (Klotho) and increased protein levels of KIM-1 (Havcr1) with immunofluorescence staining that correlate with reduced KI and increased KIM-1 mRNAs. Furthermore the secreted protein Spp1 protein levels increase in the plasma after injury as assessed by ELISA (Sup Fig 3c). Slc34a1 shows no change on protein expression level by immunofluorescence staining although the overall positive area of Slc34a1 is below 0.5%. As suggested by the Reviewer the long half-life of Slc34a1 likely explains the disparity between mRNA and protein level.

Suppl. Figure S3. Immunofluorescence staining of selected enhancer and super-enhancer associated proteins. (a) Representative immunostaining of SLC34A1, KL, SPP1, HNF1B and KIM-1 in kidney cortex in SHAM (n=4) and IRI (n=6) groups at day 2 after injury and (b) quantified percentage of positively stained area (SLC34A1, KL, SPP1, HNF1B, KIM-1) or cells (HNF1B). (c) ELISA of serum SPP1 concentration in SHAM and IRI groups (n=7) at day 2 after injury. ** P < 0.01, **** P < 0.0001. Data represent the mean \pm SD. Scale bar: 50 μ m

2) The title should indicate ischemic acute kidney injury and not acute kidney injury, as authors indicate In the text, there are multiple causes of AKI and in this manuscript authors only focus on ischemia, perhaps other inducers show different dynamics than the ones observed in this manuscript

We changed the title to: 'Enhancer and Super-Enhancer Dynamics in Repair after Ischemic Acute Kidney Injury'

3) Please define BRD4 in the abstract and FDR in the text

BRD4 and FDR are now defined in the abstract and text.

4) The UUO data with JQ1 is definitely not novel and should identify in the text that the data is similar to that from previous studies as in references 62 and 63 of the manuscript. These references already show previous evidence of JQ1 protective effects in fibrosis in the UUO model, so figure 7A is not providing more data beyond what is already published.

We indicated that our UUO data demonstrate the reproducibility of already published studies assessing the effect of JQ1 in UUO. Please see page 16 in the manuscript.

Reviewer #3 (Remarks to the Author):

In order to determine transcriptional regulators of the repair response after acute kidney injury, the authors of this manuscript profiled enhancer and super enhancer repertoire in uninjured and repairing kidneys on day two following ischemia reperfusion injury. They identified several transcription factor binding sites on enhancer and super enhancer regions. In order to establish role of BRD4 on the super enhancer function authors have used BET inhibitor JQ1. They show that JQ1 (unexpectedly) increased mortality at day two and three after ischemia reperfusion injury compared to pre ischemia reperfusion injury. On the other hand, the found JQ1 can block kidney fibrosis in unilateral ureter obstruction and aristolochic acid kidney injury models.

Altogether authors nicely showed the role of BET proteins in this AKI study using a relevant mice model system. Overall the manuscript is well written and conclusions have translational significance especially for needed caution while evaluating future BET inhibitor therapies for AKI.

The in vivo studies are good. On the other hand, authors need to do a more detailed characterization of the enhancers and super-enhancers in their model system, esp as this is the

more novel part. Moreover more experiments are need to demonstrate the connection between the enhancer profiling and the JQ1 interventions. Specific comments, suggestions and points to be addressed to further strengthen their manuscript are detailed below.

We appreciate the positive comments of the Reviewer with respect to our demonstration of the role of BET proteins in our in vivo studies.

1. Authors need to experimentally validate some of the differential enhancers in SHARED, IRI decreased and IRI increased groups by performing ChIP with H3K27ac, BRD4 and Pol II antibody at candidate target and control regions.

We performed all our ChIP-seq experiments in biological replicates with high correlation coefficients between the replicates (Supplemental material). All ChIP-seq profiles are publicly available on the Gene Expression Omnibus data repository. Any genomic region or gene of interest can be looked up in both replicates for H3K27ac, BRD4, H3K4me3, Pol II, HNF4A, GR, STAT3 and STAT5.

The link to the GEO dataset can be found on the title page or below:

To review GEO accession GSE114294:

Go to: <https://www.ncbi.nlm.nih.gov/geo/query/acc.cgi?acc=GSE114294>

Enter token **ehmhuyygtfoblyl** into the box

Besides the sequencing raw data we also provided the bedgraph files for visualization of the ChIP-seq tracks in common genome browser applications (eg. <https://genome.ucsc.edu/>) for the reader.

2. There should be some experimental connection between the first part of the paper and second part (JQ1 treatment). The authors should show whether some of the critical enhancers and superenhancers that are altered in the AKI models and differential expression their putative associated genes are differentially affected by JQ1 treatment (using samples from the JQ treated mice). Comparison of such factors during early IRI (inflammation) and in the fibrosis models could give valuable clues underlying the differential response to JQ1 and the associated genes in the AKI versus fibrosis models. This in turn could potentially help design better BET inhibitors for AKI.

We thank the reviewer for this comment. JQ1 inhibits the two extraterminal bromodomains of the BET family which leads to disturbed transcriptional elongation and genome-wide Pol II pausing^{22,36}. To add a clearer connection between the first part and JQ1 treatment we analyzed our enhancer repertoire in JQ1 treated mice. 73% of down-regulated genes (fold change > 2) after JQ1 treatment can be assigned to active enhancer elements (Chi-square: p<0.001). Further we performed Pol II ChIP-seq experiments with JQ1 treated kidney cortex samples and

could show a genome-wide Pol II pausing at the gene body after JQ1 treatment (Figure 6i-k). As JQ1 has this very broad effect on gene expression activity more targeted approaches on individual transcription factors would open new therapeutic avenues in the treatment of AKI and CKD. We hope the Reviewer will agree that this is beyond the scope of this first manuscript.

3. Figure 2: i.e. Identification of the transcription factor on the enhancers. This Figure can be part of some other main/supplemental figure. Although the data is interesting, it does not need an independent main figure. Authors need to perform ChIP/ChIP-seq with SHARED (HNF4a) and IRI decreased and increased (any bZIP family) transcription factors and show their specific binding on the enhancer regions. Are these transcription factors differentially regulated in SHAM and IRI treated kidney samples? Are the identified transcription factors specific for enhancer binding or they are also enriched in promoters of differentially regulated genes?

Indeed we found genome-wide enhancer specific transcription factor binding of the predicted transcription factors before and after kidney injury. We performed ChIP-seq for HNF4A, GR, STAT3 and STAT5 and could show decreased binding of HNF4A and GR at decreased enhancer sites and increased binding of STAT3 at IRI increased enhancer sites (Figure 4 and Suppl. Figure S6). We could also observe transcription factor binding at promoter sites as the 3D chromatin conformation would suggest (Figure 4 and Suppl. Figure S6). Interestingly many of the predicted transcription factors are also differentially regulated at the gene expression level after injury (Suppl. Figure S5c). A nice example is the Junb locus with additional STAT3 binding after injury (Figure 4). Junb is also up-regulated after injury. Taken together, our comprehensive enhancer and transcription factor characterization will be a very useful resource for the research community and provides a better understanding of the endogenous repair processes in the kidney.

4. Many or all the important enhancer-associated genes should to be validated experimentally (example Figure 3F).

5. It is important to show that these enhancers are functional. I suggest performing luciferase assays by using the enhancer constructs (For example 3g, 3h and 3l).

6. Authors can mutate or delete the super-enhancer or constituent enhancer regions of SPP1 and HNF1b and show these super-enhancers are specific to their associated genes.

We thank the reviewer for these critical comments but hope that he/she will not require the massive amount of experiments that would be required to appropriately respond to these suggestions. Functional validation of enhancer activity is of course of interest. High throughput screens have been used to examine enhancer activity in *in-vitro* models³⁹. Such assays combine next generation sequencing with massively parallel reporter assays for quantitative analysis of

transcriptional activity of thousands of enhancers in cell models, but do not allow for the identification of target genes of the assessed enhancers by using minimal or general promoter elements or to assess the combinatorial regulation of multiple enhancers. Further the cell models lack the natural chromatin context such as active chromatin state of kidney epithelia cells *in-vivo*, transcription factor regulation and transcriptional activity. Therefore, functional enhancer validation of *in-vivo* enhancer elements can be only performed by *in-vivo* experiment with targeted enhancer deletions in mice, which is beyond the scope of this study. A recent paper assessed the effects of enhancer and promoter elements on transcriptional burst kinetics⁴⁰. In their analysis it seems a high proportion of enhancers are regulating the nearest/next gene, which justifies our approach for the integration of the ChIP-seq and RNA-seq data in Figure 2 on a genome-wide scale. Further we could show specific transcription factor binding at enhancer elements adding evidence to the functionality of the identified enhancer elements in kidney epithelia cells *in-vivo*. However, specific functional enhancer, super-enhancer and promoter interactions of loci of interest must be validated in CRISPR mice. In case of Spp1 8 CRISPR mouse lines for each of the 8 single enhancers and 247 possible enhancer deletion combinations (deletion of more than one enhancer at the same time) would be necessary to fully dissect the functionality and hierarchy of this super-enhancer.

7. Since JQ1 can affect other BET proteins besides Brd4, Did authors check the expression of candidate BET proteins in SHAM and IRI treated kidney samples? Can authors comment on the role of other BET proteins while demonstrating JQ1 role in transcriptional kidney repair programs?

We conducted additional experiments to address this suggestion of the Reviewer. BRD4, BRD3 and BRD2 are not significantly regulated after IRI (48 hours after injury). We do see a trend of up-regulation of BRD4 on gene expression level after IRI, but without reaching significance (unadjusted p-value = 0.09). We also performed immunofluorescence staining for BRD4, BRD3 and BRD2. At the protein level BRD4 is the most abundant member of the BET family in the kidney before and after IRI (Suppl. Figure S7a). Further we performed ChIP-seq for BRD3 and BRD2 and compared these result to BRD4 ChIP-seq profiles. BRD4 shows the highest binding profile (coverage) compared to BRD2 with some coverage and BRD3 hardly any coverage on a genome-wide scale. Therefore our data suggest BRD4 is the dominant BET protein in kidney epithelia cells (Suppl. Figure S7b).

9. In the methods section, please provide sonication conditions and antibody concentrations used for ChIP.

As suggested we added sonication conditions and antibody concentrations used for ChIP in the Methods sections. We used 22 minutes of sonication with a probe sonicator on ice and 10µg of antibody in all ChIP-seq experiments.

References

- 1 Winter, G. E. *et al.* BET Bromodomain Proteins Function as Master Transcription Elongation Factors Independent of CDK9 Recruitment. *Mol Cell* **67**, 5-18.e19, doi:10.1016/j.molcel.2017.06.004 (2017).
- 2 Yang, L. *et al.* KIM-1-mediated phagocytosis reduces acute injury to the kidney. *J Clin Invest* **125**, 1620-1636, doi:10.1172/jci75417 (2015).
- 3 Kaminski, M. M. *et al.* Direct reprogramming of fibroblasts into renal tubular epithelial cells by defined transcription factors. *Nat Cell Biol* **18**, 1269-1280, doi:10.1038/ncb3437 (2016).
- 4 Zager, R. A. & Johnson, A. C. M. Acute kidney injury induces dramatic p21 upregulation via a novel, glucocorticoid-activated, pathway. *Am J Physiol Renal Physiol* **316**, F674-f681, doi:10.1152/ajprenal.00571.2018 (2019).
- 5 Chen, J., Chen, J. K., Conway, E. M. & Harris, R. C. Survivin mediates renal proximal tubule recovery from AKI. *J Am Soc Nephrol* **24**, 2023-2033, doi:10.1681/asn.2013010076 (2013).
- 6 Cook, H. T. *et al.* AP-1 transcription factor JunD confers protection from accelerated nephrotoxic nephritis and control podocyte-specific Vegfa expression. *Am J Pathol* **179**, 134-140, doi:10.1016/j.ajpath.2011.03.006 (2011).
- 7 Lu, C. *et al.* EGF-recruited JunD/c-fos complexes activate CD2AP gene promoter and suppress apoptosis in renal tubular epithelial cells. *Gene* **433**, 56-64, doi:10.1016/j.gene.2008.11.015 (2009).
- 8 Pillebout, E. *et al.* JunD protects against chronic kidney disease by regulating paracrine mitogens. *J Clin Invest* **112**, 843-852, doi:10.1172/jci17647 (2003).
- 9 Ichimura, T., Hung, C. C., Yang, S. A., Stevens, J. L. & Bonventre, J. V. Kidney injury molecule-1: a tissue and urinary biomarker for nephrotoxicant-induced renal injury. *Am J Physiol Renal Physiol* **286**, F552-563, doi:10.1152/ajprenal.00285.2002 (2004).
- 10 Mishra, J. *et al.* Identification of neutrophil gelatinase-associated lipocalin as a novel early urinary biomarker for ischemic renal injury. *J Am Soc Nephrol* **14**, 2534-2543, doi:10.1097/01.asn.0000088027.54400.c6 (2003).
- 11 Xie, Y. *et al.* Expression of osteopontin in gentamicin-induced acute tubular necrosis and its recovery process. *Kidney Int* **59**, 959-974, doi:10.1046/j.1523-1755.2001.059003959.x (2001).
- 12 Noiri, E. *et al.* Reduced tolerance to acute renal ischemia in mice with a targeted disruption of the osteopontin gene. *Kidney Int* **56**, 74-82, doi:10.1046/j.1523-1755.1999.00526.x (1999).
- 13 Persy, V. P., Verhulst, A., Ysebaert, D. K., De Greef, K. E. & De Broe, M. E. Reduced postischemic macrophage infiltration and interstitial fibrosis in osteopontin knockout mice. *Kidney Int* **63**, 543-553, doi:10.1046/j.1523-1755.2003.00767.x (2003).
- 14 Kumar, S., Liu, J. & McMahon, A. P. Defining the acute kidney injury and repair transcriptome. *Seminars in nephrology* **34**, 404-417, doi:10.1016/j.semnephrol.2014.06.007 (2014).
- 15 Liu, J. *et al.* Molecular characterization of the transition from acute to chronic kidney injury following ischemia/reperfusion. *JCI Insight* **2**, doi:10.1172/jci.insight.94716 (2017).
- 16 Pavkovic, M. *et al.* Multi omics analysis of fibrotic kidneys in two mouse models. *Scientific data* **6**, 92, doi:10.1038/s41597-019-0095-5 (2019).
- 17 Su, Z. *et al.* Comparing next-generation sequencing and microarray technologies in a toxicological study of the effects of aristolochic acid on rat kidneys. *Chemical research in toxicology* **24**, 1486-1493, doi:10.1021/tx200103b (2011).

- 18 Canaud, G. & Bonventre, J. V. Cell cycle arrest and the evolution of chronic kidney disease from acute kidney injury. *Nephrol Dial Transplant* **30**, 575-583, doi:10.1093/ndt/gfu230 (2015).
- 19 Yang, L., Humphreys, B. D. & Bonventre, J. V. Pathophysiology of acute kidney injury to chronic kidney disease: maladaptive repair. *Contrib Nephrol* **174**, 149-155, doi:10.1159/000329385 (2011).
- 20 Cippa, P. E. *et al.* Transcriptional trajectories of human kidney injury progression. *JCI Insight* **3**, doi:10.1172/jci.insight.123151 (2018).
- 21 Bonev, B. & Cavalli, G. Organization and function of the 3D genome. *Nature reviews. Genetics* **17**, 661-678, doi:10.1038/nrg.2016.112 (2016).
- 22 Filippakopoulos, P. *et al.* Selective inhibition of BET bromodomains. *Nature* **468**, 1067-1073, doi:10.1038/nature09504 (2010).
- 23 De Raedt, T. *et al.* PRC2 loss amplifies Ras-driven transcription and confers sensitivity to BRD4-based therapies. *Nature* **514**, 247-251, doi:10.1038/nature13561 (2014).
- 24 Delmore, J. E. *et al.* BET bromodomain inhibition as a therapeutic strategy to target c-Myc. *Cell* **146**, 904-917, doi:10.1016/j.cell.2011.08.017 (2011).
- 25 Lee, J. E. *et al.* Brd4 binds to active enhancers to control cell identity gene induction in adipogenesis and myogenesis. *Nat Commun* **8**, 2217, doi:10.1038/s41467-017-02403-5 (2017).
- 26 Loven, J. *et al.* Selective inhibition of tumor oncogenes by disruption of super-enhancers. *Cell* **153**, 320-334, doi:10.1016/j.cell.2013.03.036 (2013).
- 27 Mertz, J. A. *et al.* Targeting MYC dependence in cancer by inhibiting BET bromodomains. *Proc Natl Acad Sci U S A* **108**, 16669-16674, doi:10.1073/pnas.1108190108 (2011).
- 28 Najafova, Z. *et al.* BRD4 localization to lineage-specific enhancers is associated with a distinct transcription factor repertoire. *Nucleic Acids Res* **45**, 127-141, doi:10.1093/nar/gkw826 (2017).
- 29 Shi, J. *et al.* Disrupting the interaction of BRD4 with diacetylated Twist suppresses tumorigenesis in basal-like breast cancer. *Cancer Cell* **25**, 210-225, doi:10.1016/j.ccr.2014.01.028 (2014).
- 30 Xiong, C. *et al.* Pharmacological targeting of BET proteins inhibits renal fibroblast activation and alleviates renal fibrosis. *Oncotarget* **7**, 69291-69308, doi:10.18632/oncotarget.12498 (2016).
- 31 Zhou, B. *et al.* Brd4 inhibition attenuates unilateral ureteral obstruction-induced fibrosis by blocking TGF-beta-mediated Nox4 expression. *Redox Biol* **11**, 390-402, doi:10.1016/j.redox.2016.12.031 (2017).
- 32 Zhou, X. *et al.* Therapeutic targeting of BET bromodomain protein, Brd4, delays cyst growth in ADPKD. *Hum Mol Genet* **24**, 3982-3993, doi:10.1093/hmg/ddv136 (2015).
- 33 Zuber, J. *et al.* RNAi screen identifies Brd4 as a therapeutic target in acute myeloid leukaemia. *Nature* **478**, 524-528, doi:10.1038/nature10334 (2011).
- 34 Brown, J. D. *et al.* NF-kappaB directs dynamic super enhancer formation in inflammation and atherogenesis. *Mol Cell* **56**, 219-231, doi:10.1016/j.molcel.2014.08.024 (2014).
- 35 Shi, J. & Vakoc, C. R. The mechanisms behind the therapeutic activity of BET bromodomain inhibition. *Mol Cell* **54**, 728-736, doi:10.1016/j.molcel.2014.05.016 (2014).
- 36 Kanno, T. *et al.* BRD4 assists elongation of both coding and enhancer RNAs by interacting with acetylated histones. *Nature structural & molecular biology* **21**, 1047-1057, doi:10.1038/nsmb.2912 (2014).
- 37 Park, K. M., Chen, A. & Bonventre, J. V. Prevention of kidney ischemia/reperfusion-induced functional injury and JNK, p38, and MAPK kinase activation by remote ischemic pretreatment. *J Biol Chem* **276**, 11870-11876, doi:10.1074/jbc.M007518200 (2001).
- 38 Bolden, J. E. *et al.* Inducible in vivo silencing of Brd4 identifies potential toxicities of sustained BET protein inhibition. *Cell reports* **8**, 1919-1929, doi:10.1016/j.celrep.2014.08.025 (2014).
- 39 Arnold, C. D. *et al.* Genome-wide quantitative enhancer activity maps identified by STARR-seq. *Science* **339**, 1074-1077, doi:10.1126/science.1232542 (2013).

40 Larsson, A. J. M. *et al.* Genomic encoding of transcriptional burst kinetics. *Nature* **565**, 251-254, doi:10.1038/s41586-018-0836-1 (2019).

REVIEWERS' COMMENTS:

Reviewer #1 (Remarks to the Author):

The authors have done a good job in addressing my comments. The manuscript provides interesting insights into the response to injury in the kidney. It has recently been reported that drugs targeting BD1 and BD2 may have a different effect on altering the response to injury and inflammation (Gilan et al; Science 2020) and it would be useful for the readers to have the authors discuss the implications of selective bromodomain inhibition of the BET proteins in the context of their findings. I have no further experimental suggestions and congratulate the authors on their study.

Reviewer #2 (Remarks to the Author):

The authors made a remarkable job addressing the issues I had. I have no more concerns.

Reviewer #3 (Remarks to the Author):

Authors have provided an improved version of the manuscript and addressed most of my concerns. In the revised version, authors have provided new ChIP-seq data for transcription factors HNF4A, GR, STAT3 and STAT5 in kidney epithelial cells in order to establish the role of these transcription factors on enhancers and super-enhancers, which is an important piece of information.

However, I have some minor comments to be addressed.

On Fig 2. g, h, and i, please show the RNA seq track for Slc34a1, Kl and Havcr1 and highlight the enhancer regions. This will make it easier to understand how enhancers are affecting the expression of nearby genes.

Similarly, for figure 3. e and f, display RNA-seq tracks and highlight the super enhancer regions for Spp1 and Hnf1b. Enhancer for Havcr1 does not have good signal for H3K27ac (Fig 2i).

We wish to thank each of the Reviewers for the care with which they have reviewed our manuscript and for providing so many useful suggestions.

Reviewers' comments:

Reviewer #1 (Remarks to the Author):

The authors have done a good job in addressing my comments. The manuscript provides interesting insights into the response to injury in the kidney. It has recently been reported that drugs targeting BD1 and BD2 may have a different effect on altering the response to injury and inflammation (Gilan et al; Science 2020) and it would be useful for the readers to have the authors discuss the implications of selective bromodomain inhibition of the BET proteins in the context of their findings. I have no further experimental suggestions and congratulate the authors on their study.

We added this interest point of the different functionality of BD1 and BD2 on BET proteins in the discussion of our manuscript (page 19 and 20).

Reviewer #2 (Remarks to the Author):

The authors made a remarkable job addressing the issues I had. I have no more concerns.

We thank reviewer 2 for reviewing our manuscript and his useful comments.

Reviewer #3 (Remarks to the Author):

Authors have provided an improved version of the manuscript and addressed most of my concerns. In the revised version, authors have provided new ChIP-seq data for transcription factors HNF4A, GR, STAT3 and STAT5 in kidney epithelial cells in order to establish the role of these transcription factors on enhancers and super-enhancers, which is an important piece of information.

However, I have some minor comments to be addressed.

On Fig 2. g, h, and i, please show the RNA seq track for Slc34a1, Kl and Havcr1 and highlight the enhancer regions. This will make it easier to understand how enhancers are affecting the expression of nearby genes. Similarly, for figure 3. e and f, display RNA-seq tracks and highlight the super enhancer regions for Spp1 and Hnf1b. Enhancer for Havcr1 does not have good signal for H3K27ac (Fig 2i).

RNA-seq tracks from SHAM and IRI samples were added to Fig 2 g, h and i, and Fig 3 e and f. Based on the RNA-seq tracks the expression levels of the nearby genes can be visually assessed. We thank the reviewer for his comments improving our manuscript.